The ontogenetic transformation of the mesosaurid tarsus: a contribution to the origin of the primitive amniotic astragalus

Piñeiro Graciela 1 fossil@fcien.edu.uy
Núñez Demarco Pablo 2
Meneghel Melitta D. 3
1 Instituto de Ciencias Geológicas, Facultad de Ciencias. Iguá, Universidad de la República , Montevideo , Uruguay
2 Instituto de Ciencias Geológicas, Iguá, Facultad de Ciencias , Montevideo , Uruguay
3 Laboratorio de Sistemática e Historia Natural de Vertebrados, IECA, Facultad de Ciencias, Iguá , Montevideo , Uruguay
Abdala Virginia
Electronic publication date: 2016 May 17
Publication date: 2016
Volume: 4
Electronic Location ID: e2036
Received 2015 Dec 31; Accepted 2016 Apr 22
Copyright: ©2016 Piñeiro et al.
Copyright year: 2016
Copyright holder: Piñeiro et al.
License: This is an open access article distributed under the terms of the Creative Commons Attribution License, which permits unrestricted use, distribution, reproduction and adaptation in any medium and for any purpose provided that it is properly attributed. For attribution, the original author(s), title, publication source (PeerJ) and either DOI or URL of the article must be cited.
License URL: https://creativecommons.org/licenses/by/4.0/

Keywords: Mesosaur ontogeny, Astragalus formation, Evolutionary studies, Navicular origin, Implicated groups, Amniotes, Non Amniotes

Funding: ANII-FCE 2011_6450 NGS 9497_14 This study was funded by ANII-FCE 2011_6450 and NGS Grant 9497_14 (to GP). The funders had no role in study design, data collection and analysis, decision to publish, or preparation of the manuscript.

==============================
The hypotheses about the origin of the primitive amniotic tarsus are very speculative. Early studies argued that the origin of the astragalus, one of the largest proximal bones in the tarsus of basal amniotes, was produced by either the fusion of two, three, or even four of the original tarsal bones, the intermedium, the tibiale and the proximal centralia (c4 and c3), or that the intermedium alone transforms into the primitive astragalus. More recent studies have shown that the structure of the tarsus in Captorhinus supports the former hypothesis about a fusion of the intermedium, the tibiale, the proximal centrale (c4) and eventually c3, producing a purportedly multipartite structure of the amniotic astragalus, but the issue remained contentious. Very well preserved tarsi of the Early Permian aquatic amniote Mesosaurus tenuidens Gervais, 1864–1865, which represent the most complete ontogenetic succession known for a basal amniote (the other exceptional one is provided by the Late Permian diapsid Hovasaurus boulei Piveteau, 1926), suggest that there is more than one ossification center for the astragalus and that these fuse during late embryonic stages or maybe early after birth. A non-hatched Mesosaurus in an advanced stage of development shows that the tarsus is represented by a single bone, most probably the astragalus, which seems to be formed by the suturing of three bones, here interpreted as being the intermedium, the tibiale, probably already integrated to the c4 in an earlier stage of the development, and the c3. An amniote-like tarsal structure is observed in very basal Carboniferous and Permian tetrapods such as Proterogyrinus, Gephyrostegus, the diadectids Diadectes and Orobates, some microsaurs like Tuditanus and Pantylus and possibly Westlothiana, taxa that were all considered as true amniotes in their original descriptions. Therefore, the structure of the amniotic tarsus, including the configuration of the proximal series formed by the astragalus and the calcaneum, typically a pair of enlarged bones, could have been established well before the first recognized amniote walked on Earth. Accordingly, the tarsus of these taxa does not constitute specialized convergences that appeared in unrelated groups, they might be instead, part of a transformation series that involves taxa closely related to the early amniotes as some hypotheses have suggested.

Introduction

The origin of the astragalus and the calcaneum in the ankle of basal amniotes has been considered as an adaptation to terrestrial locomotion and a key innovation in the origin of Amniota (Romer, 1956). Taking into account the elements present in the tarsus of basal tetrapods, it is clear that there was a strong reduction in the number of bones that form the primitive amniotic tarsus. This reduction can be explained by the fusion or loss of some tarsal bones in the ancestral amniotes despite the homology of these elements not always is well established. According to previous contributions, it is widely acknowledged that the calcaneum is derived from the fibulare, i.e., from only one of the precursor bones present in the tarsus of non-amniote tetrapods. However, the origin of the astragalus, as well as the identification of the ancestral bones that give origin to it, are contentious (Peabody, 1951; Rieppel, 1993; Kissel, Dilkes & Reisz, 2002; Berman & Henrici, 2003; O’Keefe et al., 2006; Meyer & Anderson, 2013). Some authors supported the classic hypothesis of a unitary origin for the astragalus, from the intermedium (see Romer, 1956) or perhaps from the fusion of this bone to the tibiale (e.g., Holmgren, 1933; Gegenbaur, 1864 in Schaeffer, 1941). However, Peabody, 1951, following Holmgren (1933), suggested that the origin of the astragalus is produced by the fusion of three bones; mainly the intermedium, one of the proximal centralia (c4) and perhaps, the tibiale (Peabody, 1951, Fig. 2). A modification of this proposal, although supporting the composite origin for the astragalus, was suggested by O’Keefe et al. (2006) by including also the third centrale as a component of the fused element (four-center hypothesis). Indeed, there is evidence of a fusion between the tibiale and the proximal centrale (c4) in Gephyrostegus (Schaeffer, 1941; Holmes, 1984) which possesses an amniote-like tarsus (Carroll, 1970), thus, this fusion may have occurred early in the evolution of the amniotic tarsus. Peabody’s (1951) hypothesis was subsequently refuted by Rieppel (1993) who stated, based on embryological evidence from extant reptiles, that the reptilian astragalus is a neomorph. But Rieppel’s (1993) suggestion was not widely accepted and the hypothesis on the multipartite structure of the reptilian astragalus remains plausible. Recent reports of well-preserved tarsi from apparently young individuals of several captorhinid species (Kissel, Dilkes & Reisz, 2002; Berman & Henrici, 2003; O’Keefe et al., 2005; O’Keefe et al., 2006), which will be discussed later, demonstrate that the matter is still open.

Embryological studies show only two cartilaginous condensations close to the distal end of the fibula in most extant reptiles, one for the astragalus and the other for the calcaneum (Schaeffer, 1941; Rieppel, 1993), but the presence of additional anlagen for the tibiale, remains contentious. Mainly due to this evidence, the widespread view about the origin of the astragalus before Peabody’s (1951) contribution was in favor of a slightly transformed intermedium as the astragalus precursor.

Another characteristic of the primitive amniotic tarsus is the articulation of the proximal tarsal elements (astragalus and calcaneum) with centralia 1 and 2, which are placed distally and often fuse to each other (Peabody, 1951). The fused element (c1 + c2), commonly named the centrale or lateral centrale, has been suggested to form the navicular bone, characteristically present in therapsid-grade synapsids and mammals (Broom, 1915; Broom, 1924; Jenkins, 1971). Moreover, five distal tarsals are present, the first and the fourth commonly being the largest.

Here we investigate the origin and evolution of the amniotic astragalus by a thorough study of several almost complete and some incomplete mesosaur skeletons and natural external molds and casts, including well-preserved feet. Moreover, well preserved, isolated astragali and calcanea of individuals in different ontogenetic stages, including the tarsus of one non-hatched Mesosaurus tenuidens and hatchling individuals, were also analyzed for completing an ontogenetic sequence previously unknown for any other Early Permian amniote. This amazing record provides useful data for characterizing the tarsal structure in early and late juvenile stages, and helps us to understand the transition towards the acquisition of the adult tarsal morphology. We present a synoptic view of the evidence we found for homologizing the primitive amniotic astragalus to the intermedium plus possibly the tibiale and proximal centralia, and propose that the suturing of these elements occurred during the embryonic stage, producing a very specialized single bone in the hatchlings. We also report the invariable presence of a navicular-like bone (fusion of c1 + c2?) in Mesosaurus tenuidens (contra Modesto, 1996a; Modesto, 1996b; Modesto, 1999) and discuss the possibility if this character is polymorphic for mesosaurs as observed in basal synapsids (Romer & Price, 1940).

Materials and Methods

The specimens used in this study are part of several palaeontological collections and consist of almost complete and well preserved Mesosaurus tenuidens individuals (Gervais, 1864–1865; Gervais, 1865) and partially preserved skeletons that include the hind limbs, which are the subject of our study. They allow us to address the structure of the mesosaur tarsus and its component bones at different stages of development. All these materials plus isolated complete astragali and calcanea from juvenile and mature individuals were analyzed by using a binocular microscope and different techniques of photography, as well as by digital drawings. Specimens from FC-DPV, GP/2E, MN and SMF-R were personally analyzed by the senior author (GP), while the specimens from the AMNH were studied from photographs kindly provided by personnel of that institution.

Methods

In order to evaluate the structure and ontogenetic variation of the mesosaurid tarsus, particularly that of the astragalus, we carried out an anatomical study of 50 mesosaurid specimens assigned to the species Mesosaurus tenuidens. We selected 18 individuals with well-preserved tarsi, including a non-hatched individual in a late stage of development, to represent an idealized ontogenetic transition (Figs. 1–6).

Figure 1 Mesosaurus tenuidens, ontogenetic transformation in the tarsus formation.

Photographs of the selected specimens preserving epipodial, mesopodial and metapodial elements. The images focussed particularly on the tarsal elements preserved in each of the specimens. This figure includes the earliest stages of the ontogenetic series. (A) FC-DPV 2504, close-up view of the limbs preserved in a non-hatched mesosaurid. The very small composite tarsus can be seen slightly distally displaced from its natural position close to the zeugopodium. See the interpretive drawings in Figs. 2A and 7 and text for further description. (B) GP-2E 272, tarsus of a very young mesosaur; the constituent elements should have already started ossification, but they are covered by the plantar aponeurosis and just shadows of astragalus and distal tarsals can be seen. See interpretive drawings in Fig. 2B for details, (C) SMF-R 4496, well preserved tarsus of a young individual, both astragalus and calcaneum can be observed close to the crus. See the interpretive drawing in Fig. 2C for a more detailed anatomical description of the specimen. (D) AMNH 23795, tarsus of a very young mesosaur showing the astragalus and a tiny calcaneum a little laterally displaced. The calcaneum still preserves part of the suturing of the precursor bones over its visible (probably ventral) surface. Toe number one is not completely ossified yet, suggesting a very juvenile stage of this specimen. See interpretive drawings in Fig. 2D for more detailed anatomical description of the specimen. (E–G) MN 4741, SMF-R 4934, and SMF-R 4513, show the progressive growing of the individuals in the ontogenetic series and the concomitant dramatic changes in the morphology of the astragalus. According to the tarsus morphology and the further ossification of the limbs and overall skeleton, the specimen in (G) is considered to be a young adult or a sub-adult. See text for further descriptions and interpretive drawing in Figs. 2E–2G.

Figure 2 Mesosaurus tenuidens, ontogenetic transformation in the tarsus formation.

Interpretive drawings of the specimens in Fig. 1. See text for further descriptions of each included specimen. Scale bar: 5 mm.

Figure 3 Mesosaurus tenuidens, ontogenetic transformation in the tarsus formation.

Detailed interpretive drawings to show the morphology of the tarsus in hatchling and juvenile mesosaurid shown in Fig. 1 (B–G; A, is detailed in Fig. 9). Putative ancestral bones that formed the mesosaur astragalus are shown as we interpreted them based on the morphology and relationships of the tarsal bones preserved in FC-DPV 2504, the non-hatched mesosaurid (see Fig. 9 and text for further descriptions of each the included specimens). Anatomical Abbreviations: ?ac3, putative ancestral centrale three; ?ai, putative ancestral intermedium; as, astragalus; ?ate + ac4, putative ancestral tibiale plus ancestral centrale four; ?c2, putative centrale two; ca, calcaneum; ?ca, possible alternative calcaneum; ?dt, putative distal tarsals; ?dt4, putative distal tarsal four; ?na, putative navicular; pa, plantar aponeurosis.

Figure 4 Mesosaurus tenuidens, ontogenetic transformation in the tarsus formation.

Photographs of the selected specimens preserving epipodial, mesopodial and metapodial elements. From (H) to (P) GP-2E 5610, FC-DPV 2497, GP-2E 114, SMF-R 4710, SMF-R 4470, GP-2E 5816, GP-2E 6576, GP-2E 5740, FC-DPV 2058. All the specimens are considered as adults; they have well ossified tarsi. The preserved bones and their morphology fit into the typical pattern for basal amniotes: 2 large proximal bones (astragalus and calcaneum), a ‘navicular’ (often preserving the suture between c1 and c2) and 5 distal tarsals. See Figs. 5 and 6 for interpretive drawings of the preserved bones and their main characteristic features.

Figure 5 Mesosaurus tenuidens, ontogenetic transformation in the tarsus formation.

Interpretive drawings of the specimens in Fig. 2 (H–P) showing the adult stages in the ontogenetic sequence. See text for further descriptions of each the included specimens.

Figure 6 Mesosaurus tenuidens, ontogenetic transformation in the tarsus formation.

Detailed interpretive drawings of the specimens in Fig. 2 showing the morphology of the tarsus in adult individuals. The formation of the ‘navicular’ by the fusion of c1 and c2 is shown through the series, as well as the formation and development of the foramen for the perforating artery. Notable is the variation in size and shape of the distal tarsals observed in the analysed specimens. Anatomical abbreviations: as, astragalus; c1, centrale 1; c2, centrale 2; ca, calcaneum; na: ‘navicular’; paf, foramen for the perforating artery; I, II, III, IV, V, distal tarsals.

Distinction of juvenile from adult mesosaurs

The recognition of young, immature individuals from adult, mature ones was not easy to determine in mesosaurs. Modesto (1996a), Modesto (1999), Modesto (2006) and Modesto (2010) made a detailed study of the characters that can be used to recognize the three monospecific genera that compose the Family Mesosauridae. He concluded that the main characters (e.g., tooth morphology, head-to-neck ratios, presacral vertebral counts, presence/absence of pachyostotic ribs and hemal arches) used for taxonomic purposes are valid to separate three monospecific mesosaurid taxa. Nevetheless, Piñeiro (2002), Piñeiro (2004) and Piñeiro (2008) revised some of the characters that have been previously used as taxonomically diagnostic and found that they could instead be ontogenetic conditions distinguishing alternatively immature and mature specimens or could even represent sexual dimorphism. Reliable characters that can be useful to differentiate juvenile (immature) from adult (mature) mesosaurid individuals can be derived from changes in the morphology and structure of the coracoid and the scapula in the shoulder girdle and the pubis in the pelvic girdle (Piñeiro, 2004). These bones are simple rounded plate-like structures in very young individuals, only acquiring the suchlike shape in adults; the coracoid develops into a roughly rectangular bone with anterior and medial convex margins (Modesto, 1996a; Modesto, 1996b; Piñeiro, 2004). The coracoid notch pierces the bone medially but is very poorly developed in young individuals. It becomes a true coracoid foramen in adults, when both bones suture and eventually fuse to form the scapulo-coracoid. These bones can fuse leaving no trace of any suture between them, even in apparently young adults, or the suture may remain visible even in large, adult individuals (Piñeiro, 2002), evidencing perhaps intraspecific or sexual variability (Piñeiro, 2004). Similar morphological changes are seen in the pubis, from being a small, plate-like rounded bone to a more kidney-shaped element that develops a pubic notch or a true obturator foramen totally enclosed by bone. Other aspects of the skeleton morphology will be part of a forthcoming paper, and will not, therefore, be discussed here. Even though the characters reviewed above are useful as complementary data to help identify the development stage in mesosaurs, the presence of well ossified carpal and tarsal bones was the most useful feature for considering maturity in mesosaurs. We consider here that an individual is mature when in the tarsus, the astragalus and the calcaneum approach each other and the foramen for the perforating artery appears between them.

Centralia and navicular nomenclature

The c1 is often named as the lateral centrale and the c2 as the medial centrale. But, when only one centralia is seen (it could result from the fusion of c1 + c2 or it could be just the c2), it is often identified as the centrale (e.g., Schaeffer, 1941; Currie, 1981; Lewis, 1964; Reisz & Fröbisch, 2014), or as the distal centrale (e.g., Carroll, 1970) or as the lateral centrale (e.g., Peabody, 1952; Modesto, 1999; Reisz & Dilkes, 2003), even though these bones are always placed medially in the tarsus, or even as the navicular (Schaeffer, 1941). Similarly, the c4 is called the proximal centrale (e.g., Kissel, Dilkes & Reisz, 2002; Berman & Henrici, 2003) or posterior centrale (Olson 1968). On the other hand, there is no stable designation for the c3 and it can be mistaken for the c4 when it is called the proximal centrale (Carroll, 1970; Holmgren, 1933) or even considered a distal centrale (Fröbisch, 2008; Hall, 2007). This lack of consensus in the literature on how to refer to specific centralia increases the confusion about the establishment of evolutionary patterns for the early amniotic tarsus. Therefore, we decided to use the following naming criterion: we refer to the bone (or fused bones) placed distally to the astragalus in the mesosaur tarsus as the ‘navicular’, and we use the name “proximal centrale” only when it cannot be determined if it is the c4 or c3.

Systematic Palaeontology

Amniota Haeckel, 1866	
Proganosauria Baur, 1889	
Mesosauridae Baur, 1889	
Mesosaurus tenuidens Gervais, 1864–1865	
Figs. 1–9	

The mesosaurid tarsus (Figs. 1–9) displays a plesiomorphic construction regarding the structures observed in other basal amniotes as Hylonomus lyelli, Paleothyris acadiana and Petrolacosaurus kansensis (Carroll, 1964; Carroll, 1969; Peabody, 1952; Reisz, 1981). It is also essentially equivalent to the tarsus of basal synapsids (Romer & Price, 1940; Romer, 1956) and it even mirrors the structure described for some microsaurs, particularly Tuditanus, and Pantylus, the embolomere Proterogyrinus, Westlothiana and Gephyrostegus (Carroll, 1968; Carroll, 1970; Carroll & Baird, 1968; Holmes, 1984; Smithson, 1989, although see also Smithson et al., 1994) (Fig. 10).

Figure 7 Photograph (A) and anatomical reconstruction (B) of the crus in an adult Mesosaurus tenuidens.

Colours indicate the identity of the different elements that form the tarsus and the crus. Scale bar: 10 mm.

Figure 8 Ontogenetic transition of the ‘navicular’ in Mesosaurus tenuidens.

(A) FC-DPV 1502, from left to right, photographs and interpretive drawings of isolated astragalus from a young individual, in dorsal, ventral and medial views respectively. The bone shows the typical square outline of immature individuals and the remains of sutures between the original anlagen more visible on its ventral surface, which appears to display a different morphology with respect to the dorsal one. Note that there are no traces of the ‘navicular’ preserved along the distal surface of the astragalus, which bears a concave margin. (B) GP-2E 5203, photograph and interpretive drawing of astragalus, calcaneum and incipient ‘navicular’ of a young individual in dorsal view. Recall on that the ‘navicular’ is already united to the astragalus by c2, being formed by c1 and c2 and the suture between them is still well visible. (C) FC-DPV 1479, photographs and interpretive drawings of an isolated astragalus from an adult individual in dorsal, ventral and medial view. Observe that the ‘navicular’ is now a single bone almost completely fused to the astragalus to produce the finally resultant adult outline. C1 has transformed into a tip-like bone and remains separated from the astragalus, but it can just be seen from the ventral view, which still features different from the dorsal one. The wide and triangular facet for articulation with the tibia can be seen from the medial view. Anatomical abbreviations: a, astragalus; ca, calcaneum; c1, centrale one; c2, centrale two; ac3, ancestral centrale three; ft, facet for the articulation of the tibia; ai, ancestral intermedium; ate + ac4, ancestral tibiale plus ancestral central four. Scale bar: 5 mm.

Description. All specimens from Uruguay were collected either in bituminous or non-bituminous shale of the Early Permian (Artinskian) Mangrullo Formation (Piñeiro, 2004; Piñeiro et al., 2012a; Piñeiro et al., 2012b); all the material coming from Brazil was collected in the correlative Iratí Formation (Santos et al., 2006). Each of the constituent tarsal elements will be described for the specimens representing the transition regarding their ontogenetic stage and the morphological changes detected:

(1) FC-DPV 2504 (Figs. 1–2A and 9). An almost complete and well preserved non-hatched Mesosaurus tenuidens from Uruguay, which is curled as if within an egg (Piñeiro et al., 2012b). It consists of an external mould of a small, still poorly ossified skeleton that suffered strong dorsoventral compression during diagenesis. This is evidenced by the displacement of the ribs and feet which are overlapping each other, as well as by the reduced three-dimensionality (suggesting strong compression) of the delicate skeleton, which represents the smallest mesosaur yet found (see Figs. 1 and 2 to better appreciate the small size of the specimen). While some of the constituent bones of the feet may not be completely ossified (considering the small size and the poor preservation of the manus), the extraordinary preservation of the specimen allowed us to reconstruct the structure of the tarsus and to describe the bones that seem to be present (Fig. 9). Both astragali are preserved, but only one of them shows the precursor bones articulated (see Fig. 9); the other was probably affected by the lateral compression that the specimen suffered during the early stages of fossilization, producing the separation of the bones. Neither one is preserved in its original anatomical position, but they were not too much displaced. Most probably, considering the curled disposition of the skeleton, the astragali dropped from their original position close to the zeugopodium to near the metatarsals when the soft tissues were decomposed. A similar displacement is observed in very young specimens of Hovasaurus boulei as figured by Caldwell (1994). The composite astragalus is shown as if it has turned itself over before reaching its final position. This was obviously favored by the presence of the enclosing egg membrane that prevented long transportation and loss of such tiny bones. Considering this taphonomic explanation, and following the anatomical disposition of the bones we interpreted the sutured bones, to be the intermedium, the tibiale (which possibly has fused to c4) and possibly the c3, confirming Peabody’s (1951) and O’Keefe et al. (2006) theory about the presence of a composite astragalus in the tarsus of early amniotes. The c4 (and maybe also c3) ossifies early in aquatic and terrestrial reptiles (Shubin & Alberch, 1986; Rieppel, 1992a; Rieppel, 1992b; Rieppel, 1993; Caldwell, 1994, among others), and the former fuses to the tibiale in Proterogyrinus scheelei (Holmes, 1984). On the other hand, c1 and c2 (=‘navicular’) may ossify very late in mesosaurs, (Figs. 4–6 and 8). Thus, taking into account the tarsal structure shown by early amniotes, and considering that mesosaurids are a very basal group, our suggested tarsal arrangement for the non-hatched mesosaurid tarsus is plausible.

The distal tarsals are no visible in the specimen. They could be still unossified judging from the fact that distal tarsals ossify later than metatarsals in amniotes and at least metatarsals II, III, IV and V were partially, or possibly completely ossified in FC-DPV 2504, but no metatarsal I, which is apparently absent (see Sheil & Portik, 2008 and references therein). Otherwise (but very improbably) due to their very small size, they would not be visible if they were displaced between the overlapping metatarsals.

(2) GP-2E 272 (Figs. 1–3B). This specimen is a well preserved very young individual from Brazil. The ribs are not as pachyostotic as can be observed in other immature specimens, but apart from that condition, the specimen does not show relevant anatomical differences to M. tenuidens. The silhouette of part of the body can be reconstructed due to the preservation of the skin. The interdigital membrane that unites the toes to the claws can be delimited as well as the robustness of the leg musculature, even in such a young individual. What could have been the plantar aponeurosis covers most of the tarsal bones (Fig. 3B). However, two elements (maybe mineralized cartilages) placed very close to the fibula are interpreted here as a possible astragalus (the largest bone) and an incipient, smaller calcaneum, which was distally displaced. It is difficult to believe that, covered by the, highly resistant plantar membrane, this tarsal bone can appear as displaced from its original anatomical position. But considering that in very early stages of development the astragalus and the calcaneum are the only bones ossified, we hypothesize that the small size of the bone and gravity combined to move it distally after the decay of flesh tissues started, particularly damaging the skin and muscle insertions. Otherwise, the calcaneum is covered by the aponeurosis and it is not visible or it is a very small fragmentary bone that is observed medially to the fibula (see Fig. 3B). It is also possible to see shadow-like structures that can be interpreted as some of the distal tarsals (e.g., dt4), which begin to ossify at very early ontogenetic stages in extant reptiles (Caldwell, 1994; Sheil & Portik, 2008). What appear to be scratch marks (according to Sedor & Costa Da-Silva, 2004) are observed close to the left foot, possibly produced by the individual before its sudden death. But these structures more likely are part of the muscle and skin that form the base of the tail, exquisitely preserved. These taphonomic features support the hypothesis that the tarsal elements, even if still cartilaginous, could have been perfectly preserved, but covered by the plantar aponeurosis, which is not frequently observed in fossil tetrapods.

(3) SMF-R 4496 (Figs. 1–3C). This specimen constitutes an external mould of a partially preserved posterior trunk and tail, with associated pelvic girdle and limbs from the Iratí Formation. This is the specimen that best shows the structure of the tarsus in immature, juvenile mesosaurids; the preserved bones might be partially ossified. The specimen is comparatively larger than the two described above; its tarsus is formed by two small roughly rounded bones, which can be homologized with the astragalus (the larger one) and the calcaneum (the smaller one), which do not meet, but lie one in front of the other and are positioned as in adult individuals. Despite its apparent general subcircular outline, the astragalus indeed shows a structure similar to that preserved in adults or sub-adult individuals, bearing thickened articulating areas and some suture lines. Although it is difficult to establish with confidence which of the original bones are involved, it is possible to suggest a putative arrangement based on the astragalus of the non-hatched mesosaurid (see Fig. 3C).

(4) AMNH 23795 (Figs. 1–3D) is an articulated, very complete skeleton of a young mesosaur, which bears a tarsus showing the same structure seen in SMF-R 4496 (probably because they are individuals of equivalent age). Both the astragalus and the calcaneum can be seen close to each other. Again, the astragalus shows the same structure as in the small, previously analysed specimens, and what appear to be sutures between component bones can be seen on the dorsal surface (see Fig. 3D).

(5) MN 4741 and SMF-R 4934 (Figs. 1–3 E–F respectively) and SMF-R 4513 (Figs. 1–3 G) from Brazil are a little larger than the specimens previously described. Even though their similar still small size, SMF-R 4513 is probably ontogenetically older judging for the tarsal features. We can see for the first time the morphological differences between both the proximal tarsal bones in the ontogenetic series, the astragalus being transformed into a more stylized and more easily recognizable element (see for instance Fig. 3G). Astragalus and calcaneum are preserved close to each other, and the foramen for the perforating artery is incipient but visible at approximately the midpoint length between these bones (see SMF-R 4513, Figs. 1–3 G). SMF-R 4513 (Figs. 1–3 G) is probably an adult or a subadult individual. There are three bones present; two proximal tarsal elements are visible, the larger one being the astragalus which features a morphology which is similar to those observed in more mature individuals (Fig. 3). It is a stout bone tending to reach the L-shaped outline characteristic of the basalmost amniotes and some other tetrapods (see the distribution and schematic morphology of the tarsal bones in Fig. 10). The foramen for the perforating artery is placed at the midlength of the lateral margin, and an intimate area of contact is being generated between astragalus and calcaneum at this point (Fig. 3G). A small bone can be seen distal to the astragalus-calcaneum contact in SMF-R 4513, which is located proximal to the distal tarsal elements, including probably the dt4. It could be the ‘navicular’ starting to ossify, which will be well developed later, in mature Mesosaurus specimens.

(6) At later stages, these bones develop a short contact through the lateral margin of the astragalus and the medial margin of the calcaneum (Figs. 4–6 H to P), so, the remaining analysed specimens (FC-DPV 2497, GP-2E 114, GP-2E 5610, SMF-R 4710, SMF-R 44 70, GP-2E 5816, GP-2E 6576, GP-2E 5740 and FC-DPV 2058 (see Figs. 4–6 H–P) represent adult individuals. Most of them possess the complete series of tarsal elements: astragalus, calcaneum and ‘navicular’, as well as five distal tarsals, where the first and the fourth are often the largest, although this can be very variable (Fig. 6).

Figure 9 Preserved tarsus in a Mesosaurus tenuidens non-hatched individual.

(A) FC.DPV 2504, a non-hatched mesosaurid in the egg, showing the two feet overlapping each other by compression. (B) SEM image of the foot of FC-DPV 2504 focusing on the tarsal area. The astragali can be seen in the center of the figure, mixed between the metatarsals. The astragalus seems to be not preserved. (C) Interpretive drawing of the tarsus including a possible identification of the preserved bones by regarding previous hypotheses about the origin of the amniotic astragalus. The composing elements of the astragalus were colored to favour identification of the isolated bones of the left foot. Anatomical abbreviations: ?i, possible intermedium; ?c4, possible central four; fi, fibula; mc, metacarpals; ?te, possible tibiale; t, tibia. Scale bar: 1 mm.

Figure 10 Tarsus structure in basal tetrapods, including amniote and non-amniote taxa.

Schematic diagram for comparing the tarsus structure in the basal tetrapods Acheloma and Greererpeton (amphibian-like tarsus) with regard to that of embolomeres and microsaurs (amniote-like tarsus) and early amniotes. Note the similar structure and construction of the microsaur tarsus with respect to the early amniote Hylonomus. See text for more details of the evolutive significance of the selected taxa. Abbreviations: as, astragalus; i, intermedium; te, tibiale; 1, 2, 3, 4, centralia; i, ii, iii, iv, v, distal tarsals. Taxa were redrawn from the following sources: Acheloma (Dilkes, 2015); Greererpeton (Godfrey, 1989); Proterogyrinus (Holmes, 1984); Gephyrostegus (Carroll, 1970); Seymouria (Berman et al., 2000); Westlothiana (Smithson, 1989; Smithson et al., 1994); Pantylus (Carroll, 1968); Tuditanus (Carroll, 1968); Diadectomorphs (Moss, 1972; Berman & Henrici, 2003); Ophiacodon and Haptodus (Romer & Price, 1940); Hylonomus (Carroll, 1964); Captorhinus (Fox & Bowman, 1966); Petrolacosaurus (Peabody, 1952; Reisz, 1981).

In summary, the mesosaur tarsus consists of two proximal bones identified as the astragalus and the calcaneum plus a single navicular-like element and five elements in the distal tarsal series (Fig. 7), resulting in 8 or 9 tarsal bones. The bones that form the ‘navicular’ may be the centralia 1 and 2 considering that c4 and c3 ossify very early in the ontogeny of other fossil and extant sauropsids, while the former are the last to become visible (Caldwell, 1994).

Results and Discussion

Following the evidence provided by the studied specimens, which notably includes the partially preserved tarsus of a non-hatched mesosaurid in an advanced stage of development, we can see the significant morphological transformation that the mesosaur astragalus experienced during ontogeny. The non-hatched Mesosaurus tenuidens found in the Early Permian of Uruguay (see Piñeiro et al., 2012a; Piñeiro et al., 2012b) is so exquisitely preserved that it allows us to describe the morphology of what we interpret to be a composite astragalus that is one millimeter in length! It possibly shows the precursors of the typical amniotic astragalus united by weak sutures (Fig. 9). The following postnatal, early stages of mesosaur ontogeny are characterized by the presence of sub-circular to roughly square small bones, mainly representing the astragalus as a single bone (and the more frequently preserved), although some young specimens still show the tripartite structure (Figs. 1–3 C–E) which is not easy to observe directly from photographs because of the very small size of the specimens. The sutures between the precursor bones in the astragalus of larger, adult individuals can often be deduced from not always well preserved features (e.g., sutures, rugose surfaces and thickened margins) (Figs. 6 and 8C).

In the early stages of development, astragalus and calcaneum seem to have been separated, as there is no evidence of contact between them. The foramen for the perforating artery is not visible; we consider both these features as useful in identifying juvenile, immature mesosaurids. At the following stage, the astragalus becomes more quadrangular in shape, approaches the calcaneum, and an incipient foramen for the perforating artery develops. At this stage, mesosaurids appear to be young adults and possibly, mature individuals, judging by the further ossification of the overall skeleton. The remaining transformations are crucial for the growth of the individuals for improving their capabilities for capturing prey and for their reproductive traits (see Ramos, 2015; Villamil et al., 2015; Piñeiro et al., 2012a). The proximal border of the astragalus in adult individuals is deep and bears an extended rectangular facet for the fibula, making an almost immobile articulation between these bones, as in basal synapsids (Romer & Price, 1940). The foramen for the perforating artery is well developed in large (mature) individuals where the notches in both bones approach each other to form a conspicuous true foramen (see Figs. 4–6 H to P). The groove for the passage of the perforating artery crosses the bone medially and proximally, where a rugose area is visible (Figs. 4 and 6). Most likely it marks the line of suture of both of the larger bones seen in the astragalus of the non-hatched mesosaurid, implicating the intermedium and the c4 + tibiale complex. Considering this hypothesis as the most probable, another line of suture located at the medial corner of the astragalus of adult individuals may correspond to the delimitation of the tibiale and includes the articular facet for the tibia at the medial margin (Figs. 6 and 8). This suture line is also seen to be continue at the medial margin, where it runs just above the articular facet for the tibia. This facet is wide and teardrop-shaped, which allows for a broad (comparatively motile) articulation with the tibia (Figs. 8A and 8C), considering the oblique angle and the short surface at which the contact is produced. It is interesting to note that the same type of articulations (and very similarly shaped facets) for the fibula and the tibia were described for the ‘pelycosaur’ tarsus, as well as the presence of a medio-ventral extension interpreted as a cartilaginous remnant of the tibiale (Romer & Price, 1940).

Limb ossification patterns

In Mesosaurus a significant delay in mesopodial ossification is noted, following the pattern observed in most aquatic tetrapods (Rieppel, 1992a; Rieppel, 1992b; Caldwell, 1994) such as Hovasaurus boulei Currie, 1981, from which we also know an almost complete ontogenetic succession in the development of the tarsus (Caldwell, 1994). Thus, long bones (propodials, epipodials and metapodials) become ossified while the mesopodials are still formed of cartilage. However, unlike in Hovasaurus, where the astragalus and the calcaneum of very young specimens are of nearly the same size, in Mesosaurus the first is clearly larger than the latter, thus supporting the hypothesis that the astragalus is the first bone to ossify in the mesosaur tarsus, arising from the suturing and later fusion of at least three bones that are evident in the non-hatched mesosaurid. Taking into account this information, along with the condition seen in Carboniferous tetrapods and the evidence provided by the non-hatched specimen, the mesosaurid tarsal ossification proceeds in the following sequence: intermedium, tibiale + centrale 4 (and c3?, see Fig. 9 and O’Keefe et al., 2006), calcaneum, distal tarsal four, the ‘navicular’ and the remaining bones (distal tarsals 3-1 and 5). The sequence of ossification of the distal tarsal bones is not clear, however.

Contrary to what seen in extant sauropsids, where the calcaneum is the first tarsal element that ossifies (Fröbisch, 2008), the fibulare (the calcaneum homologous) ossifies much later in mesosaurs and aquatic fossil diapsids; in Hovasaurus boulei it is suggested that it appears after the c4 does (after Caldwell, 1994). Thus, it may be possible that it is already present in the tarsus of the non-hatched mesosaurid (Fig. 9), but if so, it should have been very small. Considering the presence of only two bones in juvenile individuals, identified as the astragalus and the calcaneum (Figs. 1–3), it is possible that the intermedium and the tibiale (which possibly is a composite bone if it already fused to c4) fuse early in ontogeny, as some previous workers have suggested (e.g., Gegenbaur & Williston, in Schaeffer, 1941). Indeed, the tibiale fuses to c4 in Proterogyrinus, suggesting that these bones also ossify early, and this event was proposed as the first step towards the formation of the amniotic astragalus, as both these bones also fuse to the intermedium later (Holmes, 1984).

This pattern of ossification is mostly in agreement with recent discoveries in those fields of paleontology and developmental genetics looking for patterns and processes of vertebrate limb evolution (Caldwell, 2002 and references therein). Moreover, it highlights, at least in basal tetrapods, the potential conservatism of the underlying genetic controls of limb development patterns, exceptions are related to different ecological and functional adaptations (see below).

The astragalus during ontogeny

The astragalus is the largest bone in the mesosaurid tarsus, featuring an L-shaped outline in dorsal view in mature specimens (see Figs. 4 and 7).

The shape of the astragalus changes dramatically during ontogeny; mature individuals show a stout, roughly squared bone with broad articulating facets for the crus (Figs. 8A and 8C). This bone also possesses a wide, shelf-like latero-distal facet for receiving the centrale or ‘navicular’ (Figs. 6 and 7), which can be totally separated from the astragalus, or partially fused so that the free, unfused part of the bone can only be seen on the ventral surface (Fig. 8).

However, the astragalus of immature mesosaurids is a delicate, roughly rounded or maybe subquadrangular bone bearing an evident thick dorso-medial border which developed into very well defined articulating areas during growth, producing a slightly excavated central area in the dorsal margin for the fibula and a broad, medially placed almost sub-triangular surface for the tibia. These thickened margins can be seen even in very small newborn individuals (see Figs. 1–3 C–G).

In his 1993 study, Rieppel stated that the mesosaurid astragalus does not show any evidence of being a fusion of the plesiomorphically separated tarsal elements; to him all the suture-like structures (e.g., delicate grooves or thickenings) seen on the ventral surface correspond to attachments of muscles and tendons, and the medial groove delimitates the passage of the perforating artery. Even though the mesosaur astragalus of post-hatching stages does not show the tripartite structure described in Captorhinus (Peabody, 1951; Fox & Bowman, 1966; Kissel, Dilkes & Reisz, 2002 and references therein), it seems to have been derived from the junction of at least three bones, as we can deduce from the tarsus of the non-hatched mesosaurid (Fig. 9) where we interpret although with doubts, that the incipient astragalus is the only bone in the tarsus, showing suturing for the intermedium, the tibiale and maybe both the proximal centralia (c4 + c3). Actually, some of the original joints remained in some specimens, but they show a slightly different pattern from that described by Peabody (1951) because the mediodistal Y-shaped suture for intermedium, c4 and c3 is not as evident in the studied specimens (see Figs. 3, 6 and 8).

The mesosaur ‘navicular’

The ‘navicular’ is a bone present in both synapsid and sauropsid amniotes. In the latter, it is observed at least in their basalmost representatives: a ‘navicular’ is found in captorhinids, basal diapsids, some Parareptilia and Mesosauridae and in all pelycosaurs (Figs. 8 and 10). Later, it becomes a bone that is only characteristic of derived synapsids and living mammals and it is lost in crown diapsids. In mesosaurs it ossifies at a late stage (at the same time that the foramen for the perforating artery forms) and is separated from the astragalus in most individuals or abuts against the distal margin of this bone, even fusing partially with it in mature individuals (Figs. 6 and 8). That means that the presence of the ‘navicular’ in mesosaurs is indicative of maturity.

The presence of the ‘navicular’ in Mesosaurus is a novel characteristic, as all but one (Modesto, 1996a; Modesto, 1996b; Modesto, 1999) of the previous workers did not mention its presence in descriptions of the mesosaurid tarsus. Indeed, Modesto (1996a) and Modesto (1996b) described the presence of a lateral centrale only in Stereosternum and stated that this bone is never present in Mesosaurus. We have enough evidence to confirm that a transversely elongated bone is invariably present distal to the astragalus in all the analysed mature specimens—most frequently representing two sutured bones— identified as the centralia c1 and c2 present in “pelycosaurs” and other basal amniotes. As these bones suture to the astragalus in very mature individuals, as also seems to occur in Captorhinus aguti (Peabody, 1951), it becomes difficult to identify its presence in the tarsus, as probably occurred with the specimens studied by Modesto (1996a), Modesto (1996b) and Modesto (1999) assigned to Mesosaurus tenuidens. We first become aware of the presence of a ‘navicular’ in Mesosaurus from an isolated, relatively large astragalus where the fusion of c1 and c2 has not yet been completed (see Fig. 8 for more detail of this condition). It firstly appears as two sutured (but not fused) bones (Figs. 4 and 6H–6I), and there seems to be a reduction in the size of c1, which becomes a pointed medial tip that is not preserved in most individuals because of the fragility of its suture to c2 (see Figs. 3G; 8B–8C). As a result, in Mesosaurus, the ‘navicular’ strongly abuts the platform-like facet on the distal margin of the astragalus (Figs. 6P and 8).

This variable condition concerning the fusion of centralia 1 and 2 recalls that observed in ‘pelycosaurs,’ in which some species show the centralia 1 and 2 as separate bones (e.g., Ophiacodon), while others show them fused (e.g., Haptodus) (Romer & Price, 1940) (Fig. 10). It is likely that this is an ontogenetic, perhaps heterochronic condition in mesosaurs (L Gaetano & D Marjanović, pers. comm.), but this needs to be tested by analysis of more than one individual of the same species at different stages of development. For instance, the morphology of the c1 in mesosaurids is very similar to that of the putative medial centrale of Sphenacodon ferox (according to Henrici et al., 2005), and if it is repositioned medially to the lateral central we can obtain a navicular-like bone in Sphenacodon. Thus, the small size of the tarsal bones of early amniotes and the possibility that they can be displaced from their original positions, plus to the fact that the recognition of homologous bones seems to be a difficult endeavor, make it likely that the real nature of the tarsus structure in several taxa could remain obscure. Mesosaurs may provide a good opportunity to revisit and gain a better understanding of the processes that are involved in the origin and early evolution of the amniotic tarsus.

Morphological changes supporting an evolutionary transition in the origin of the amniote tarsus

Although most previous workers (e.g., Carroll, 1964; Berman & Henrici, 2003; O’Keefe et al., 2006; Meyer & Anderson, 2013, and references therein) accepted the composite origin of the astragalus following the contribution of Peabody (1951), the reappraisal of that condition and its significance performed by Rieppel (1993) introduced controversy. This last author rejected the multipartite origin of the astragalus, arguing that there was a lack of unequivocal ontogenetic evidence that would show that the bones which would form the composite astragalus are present in at least some stage of development. He rejected the proposed composite origin of the astragalus by Peabody (1951) mainly based on the fact that this bone derives from a single ossification center in extant reptiles and that, according to Sewertzoff (1908), lizards have just a single block of cartilage close to the distal end of the fibula and tibia where the calcaneum and the astragalus later ossifies. In Sphenodon punctatus, the astragalus originates by the condensation of more than one chondrogenic element, but they fuse during the embryonic stage (Rieppel, 1993), and interestingly, there are also two chondrogenic condensations distal to the fibula in pleurodiran turtles (Fabrezi et al., 2009). In Podocnemis species for instance, one is the intermedium and the other is an elongated element, postaxially placed, which is interpreted to be the tibiale + c4 (Fabrezi et al., 2009). There is also a connective connection between c4 and the intermedium in Phrynops hylarii, showing a tarsal pattern that seems to be consistent with the basic early amniote tarsal construction as suggested by mesosaurs and other basal, non-amniote taxa.

In lizards, the tarsal formation is not as clear as in turtles. Rieppel (1992a), considered that the proximal cartilage anterior to the fibulare is the astragalus, however, there are not conclusive embryological studies that show the homology of the anterior tarsal cartilages in lizards (Fabrezi, Abdala & Martínez-Oliver, 2007). The morphogenetic approach of Shubin & Alberch (1986) seems to be useful to reconstruct the skeletal morphology in lizard limbs, and then, to identify the developmental constrains that can produce deviations in some groups from the otherwise apparently conservative pattern (see Fabrezi, Abdala & Martínez-Oliver, 2007).

On the other hand, the presence of more than one cartilage condensation, apparently homologous with the ancestral tetrapod tarsals, has been recently described to be present during early embryonic stages in the development of six different orders of modern birds (Ossa-Fuentes, Mpodozis & Vargas, 2015) and also in chameleons (Diaz & Trainor, 2015). However, their homology to the earliest amniote condition is difficult to establish,when the pattern is observed in such very specialized groups. Indeed, in the above mentioned papers, Ossa-Fuentes, Mpodozis & Vargas (2015); Diaz & Trainor (2015) it is suggested that the intermedium and the tibiale (although the latter is not pretty much apparent from the figures provided by Diaz & Trainor, 2015) appear as independent ossifications at very early stages of the development. On the other hand, Ossa-Fuentes, Mpodozis & Vargas (2015) observed that in the six groups that they studied, in contrast to the most common condition in birds (i + fe), the intermedium forms a separate ossification center that later fuses to the ‘astragalus’ (sic) forming the ascending process characteristic of dinosauromorphs. Thus, the ‘astragalus’ should be the tibiale? Moreover, the pattern of ossification that Ossa-Fuentes, Mpodozis & Vargas (2015) suggest, where the fibulare is the first to ossify, followed by the putative intermedium and later by the tibiale, is very different to that currently accepted to occur in basal amniotes.

The centralia, which are considered basic components of the astragalus structure, are recognized in stem-lepidosaurs. However, these bones are not detected in dinosauromorphs and in many extant diapsids (e.g., chameleons and birds). Therefore, they must have fused to a different bone than the astragalus or disappeared during the evolution of modern sauropsids as they are not recognizable during the ontogeny of the most advanced taxa.

Selective pressures to reduce the number of tarsal bones in the sense that they are an extension of the epipodials, favour stability by strengthening the feet to drive the body forward. Thus, the acquisition of unitary, stout structures instead of several separate, delicate bones was an improvement for sustained locomotion capabilities. Therefore, we have to be cautious regarding these findings, considering the high variability shown by the chameleons’ tarsal structure, and the lack of embryological evidence in the fossil taxa for use in comparison. Therefore, as we previously mentioned, the possibility that neomorphic elements are present in such derived groups cannot be ruled out with the available data.

Indeed, there are several known examples of tetrapods, possibly stem amniotes, that allow us to deduce the steps of fusion of the tarsal bones leading to the attainment of the amniote condition. Thus, as the embryology of extant lizards suggests, the fusion of these elements in the development of the amniote ankle is produced in the embryonic stage (Fabrezi, Abdala & Martínez-Oliver, 2007) and so, it is not possible to address their original ossification centers any more (Gauthier, Kluge & Rowe, 1988). Rieppel (1993) observed that associations of tarsal bones are common in amphibians and that, while centralia 1 and 2 can be fused or separated, c3 and c4 may be fused, or rather, one of them can be lost. Thus, according to Rieppel (1993) the association between the tibiale and c4 may be casual and may not represent a condition of phylogenetic relevance. However, we can see a real transition from closely related, supposedly non amniote taxa (e.g., Gephyrostegus, Westlothiana, Tuditanus, Pantylus (see Ruta, Coates & Quicke (2003) and Marjanović & Laurin (2015), for the phylogenetic position of these taxa), to the acquisition of the primitive amniotic tarsal configuration (see Fig. 10). Thus, if we consider the association of the tibiale and c4 observed in some Proterogyrinus specimens (Holmes, 1984) and possibly present in the tarsus of the non-hatched mesosaurid (see Fig. 9) as the first step towards the development of the amniotic tarsus (Holmes, 1984), we can reconstruct the succession including Gephyrostegus (see Carroll, 1970 as a reference of the tarsal structure in this last taxon) where the tibiale + c4 (and c3?, see O’Keefe et al., 2006) complex is associated with the intermedium to form the composite amniotic astragalus, a configuration that is also present in some microsaurs (e.g., Tuditanus punctulatus, Carroll & Baird, 1972; Carroll & Gaskill, 1978 and Pantylus cordatus, Carroll, 1968) and possibly in Westlothiana (Smithson, 1989 but see Smithson et al., 1994). Within that transformation, the fibulare becomes the calcaneum and c1 and c2 remain as the only centralia present, either as separated bones or fused to form a single element, the ‘navicular.’

Phylogenetic context supporting the evolutionary transition

On a phylogenetic point of view, even considering that there is not complete consensus about the relationships of the taxa involved in the transition, their relationships seem to be supported by the most recent cladistics analyses of basal tetrapods: Ruta, Coates & Quicke (2003); Vallin & Laurin (2004); Klembara (2005); Ruta & Coates (2007); Marjanović & Laurin (2009); Marjanović & Laurin (2015) (see Fig. 11). These phylogenies show Proterogyrinus as an embolomere anthracosaur, although the relationships of this taxon are contentious and were not completely resolved (see Ruta, Coates & Quicke, 2003). Gephyrostegus is very close to Seymouriamorpha and to microsaurs, a hypothesis supported by the Laurin & Reisz (1997) tree, which also argues that lepospondyls are a monophyletic group closely related to amniotes (see also Marjanović & Laurin, 2015). Otherwise, if microsaurs are paraphyletic to other lepospondyls and to the amniote stem, as other workers suggest (Olori, 2015), they could have been the last phylogenetic intermediaries in our evolutionary transformation series.

Figure 11 Schematic representation of recent phylogenetic hypotheses of early tetrapod relationships showing the position of the taxa involved in the evolutionary transition to the formation of the early amniotic astragalus (see text for the figure context).

(A) Ruta & Coates (2007); (B) Carroll (1995); (C) Laurin & Reisz (1999); (D) Marjanović & Laurin (2015).

It is noteworthy that some taxa which are not classified as amniotes have an amniote-like tarsus or at least developed the large proximal tarsal bones that characterize the amniotic tarsus, the astragalus and the calcaneum (Fig. 10). Notable examples of this feature are the diadectids (Romer & Byrne, 1931; Romer, 1944), although adults show the autapomorphic condition of a fusion between both the proximal bones to produce an astragalocalcaneum bone (see below). Within lepospondyls, the microsaurs Pantylus (Carroll, 1968) and Tuditanus punctulatus have intriguingly, an amniote-like tarsus (Carroll & Baird, 1972). Moreover, the proterogyrinid Proterogyrinus scheelei, Gephyrostegus bohemicus and probably Westlothiana lizziae also have an amniote-like tarsus (see Holmes, 1984; Smithson, 1989). Because mesosaurids are very basal amniotes (Laurin & Reisz, 1995; Piñeiro et al., 2012b) or basal parareptiles (Modesto, 1996a; Modesto, 1996b; Modesto, 1999; Piñeiro, 2004) we explored these taxa in order to find homologies between putative plesiomorphic, non-amniotic tarsi and their corresponding structure in mesosaurids according to the different ontogenetic stages described for the group.

The status of Westlothiana and microsaurs and its role in the transition

Regarding the condition in Westlothiana, Smithson (1989), reconstructed the tarsus as very amniote-like, including within it nine bones (see Smithson, 1989, Fig. 2D). There were certainly nine bones in the preserved material although they were not preserved in their original anatomical position. But, later, Smithson et al. (1994) pointed out that the tarsus of Westlothiana is indeed very plesiomorphic (or amphibian-like) because it included ten, rather than nine bones (see Fig. 20A in Smithson et al., 1994). We do not find enough evidence to refute the former reconstruction or for validate the latter, thus, a proposal about the tarsus structure in Westlothiana would be very speculative at this stage. Moreover, the renaming of the two large, proximally placed bones originally described as the astragalus and the calcaneum as an intermedium and a fibulare, is also speculative because this last bone is difficult to identify from the preserved specimen, where the foot bones are mostly disarticulated and obscured by the caudal vertebrae (Smithson et al., 1994). Besides, according to these authors, the putative intermedium is L-shaped, a characteristic very frequently found in the astragalus of early amniotes. Despite Westlothiana possessing other advanced conditions that may suggest its relation to the amniote clade, it also retains some plesiomorphic features in the skeleton such as a prefrontal-postfrontal contact, excluding the frontal from the orbital margin (Smithson, 1989). Thus, the reconstruction of the real structure of the tarsus in Westlothiana may be crucial to an understanding of the evolutionary transition to the origin of the amniotic astragalus as we have figured it out in this contribution. We hope that our paper will encourage new studies on this taxon.

Concerning microsaurs, these ecologically diverse, small-bodied tetrapods are credible candidates for being part of the stem leading to the emergence of the earliest amniotes. They develop a tarsus with a very amniote-like morphology, and as was recently demonstrated they even show a similar ossification pattern, with the intermedium (?astragalus) and the fibulare (?calcaneum) being the first tarsal bones to ossify (see Olori, 2015). They are also the only proximal elements in the tarsus as in all amniotes, and naming them as intermedium and fibulare is just arbitrary at this stage, if we have no embryological information to prove their identity. We have to take into account that in mesosaurids the astragalus and the calcaneum are the only proximal tarsal bones in born individuals, despite the former deriving from the fusion of three or four bones.

Diadectids

Diadectids were recently considered to be amniotes (Berman, 2000), and as such, they would have had an amniote tarsus. Recent discoveries of possible juvenile diadectid tarsi including a putative composite astragalus formed by the intermedium, the tibiale and the proximal centrale (c4, as it was identified) introduced interesting new data to the origin of the amniotic astragalus (Berman & Henrici, 2003). Later, this material was assigned to the species Orobates pabsti, a diadectid (Berman et al., 2004). Nevertheless, the holotype specimen of Orobates described by Berman & Henrici (2003) and Berman et al. (2004: 29) as having a tripartite astragalus (MNG 10181) was recently subjected to an in-deep study using micro-focus computed tomography scans (Nyakatura et al., 2015), which allowed for a thoughtful anatomical understanding of the specimen. The scanned image and digital reconstruction show that there are seven separated bones in the tarsus of Orobates, whose morphology suggests that could be homologized with immature astragalus and calcaneum plus two centralia (c1 + c2) and three small distal tarsals. Indeed, despite the very good preservation of the individual, it was apparently subjected to severe diagenetic distortion; the bones were embedded in a crystalline calcite matrix and there was a significant chemical substitution around their margins (cf. Nyakatura et al., 2015). That taphonomic feature could have produced a configuration that, under direct examination, led to the interpretation of Berman & Henrici (2003) about the presence of a composite astragalus in Orobates.

Berman & Henrici (2003) also described two associated (maybe sutured) tarsal bones which they recognized as the intermedium and the fibulare of a juvenile Diadectes. However, the shape of the bones, mostly subcircular, and their relative size and proportions, remind us of the astragalus and calcaneum of a very young individual, taking into account the ontogenetic stages described here for the very basal amniote Mesosaurus tenuidens.

This new configuration matches the pattern of the tarsus already known for diadectids: distinct astragalus and calcaneum in young individuals, which fuse later to produce an astragalocalcaneum in very mature adults. Thus, diadectids have a very amniote-like tarsus. The non-diadectid diadectomorphs (Tseajaia campi) do not have a well-defined tarsus, but this can be masked by the not sufficiently good preservation of the specimen feet. Even though, in Tseajaia campi, three distinct bones seem to form the proximal line (Moss, 1972), some fusions tending to achieve the amniote-like pattern can be hypothesized to be present: the tibiale fuses to c4 as the evolutionary transition reviewed above suggests, and the intermedium, shown by Moss (1972) as fusing to c4, indeed fuses to c3 (see Figs. 10 and 12), supporting the putative incorporation of both centralia into the amniotic astragalus as O’Keefe et al. (2006) have suggested and as it is shown by the tarsus in the non-hatched mesosaurid (Fig. 9).

Figure 12 Hypotheses about the astragalus and the navicular formation.

The schematic diagram shows the steps that lead to the formation of the amniotic tarsus, regarding the series of possible transformations that could have produced the primitive astragalus (A) as well as those that prevailed into the evolution of the ‘navicular’ bone (B).

Some groups like diadectids and seymouriamorphs for instance, show a high plasticity in producing different patterns often correlated to a different expression of otherwise highly conserved regulatory genes (Shubin, 2002). Therefore, the expression of these genes and the consecutive structure of the tarsus may be regulated by the different ecological pressures to what some have to adapt along the different stages of their development. Juvenile or young adult Diadectes show a conservative tarsus, and distinct astragalus and calcaneum were described as being present (Romer & Byrne, 1931; Romer, 1944; Berman & Henrici, 2003). However, astragalocalcaneum fusion is shown to occur in very large and mature individuals, where it would seem that the movement between these bones becomes very limited or null (Romer, 1944).

Hylonomus lyelli

Revising the evidence from other basal amniotes such as Hylonomus lyelli (Carroll, 1964; Meyer & Anderson, 2013) we found some inconsistencies related to the identification of the bones figured, perhaps as an attempt to follow the Peabody’s (1951) suggestion of a tripartite origin of the astragalus. Thus, Meyer & Anderson (2013), following Carroll (1964, Fig. 1), identified the astragalus and calcaneum from a partially disarticulated specimen where the feet are completely disassociated and considered the calcaneum of Hylonomus as two times larger than the astragalus. According to the information found in Carroll (1964, p. 72, Fig. 8) and based on the ontogenetic succession that we described here for mesosaurs, the calcaneum can sometimes be equal in size to the astragalus or even a little larger, but never that much larger. Thus, we could deduce both that it is an incomplete astragalus missing the intermedium, as Meyer & Anderson proposed in the text and in their Fig. 3 (but this would suggest that the type specimen of Hylonomus lyelli belonged to a very young individual and it does not appear to be the case, see Fig. 1 of Carroll, 1964), or that the bone identified as the calcaneum is the astragalus or that the bone is neither the astragalus nor the calcaneum. We are inclined to accept the last hypothesis because the overall small size of the individual suggests that these bones are much too large to be tarsal bones; they seem to be elements of the pelvic girdle, possibly the pubis (see Fig. 1 of Carroll, 1964). The well identified astragalus of Hylonomus lyelli (see Fig. 8 of Carroll, 1964) does not show any trace of sutures.

Captorhinids

Taking into account the previous evolutionary transition in favor of a composite origin of the amniotic astragalus, which of course may also include other taxa, the interpretation of Peabody (1951) and later workers of the presence of more than one ossification center in the astragalus of Captorhinus and other basal amniotes seems sensible. However, other extensive descriptions of Captorhinus (e.g., Fox & Bowman, 1966) do not provide more conclusive evidence about the structure of the tarsus and, as Rieppel (1993) claimed, it is necessary to have ontogenetic evidence (e.g., articulated skeletons of very young individuals of Captorhinus and/or of related taxa) to demonstrate the homology of the bones composing the tripartite astragalus and their presence in the earliest stages of development. Isolated astragali from the Lower Permian of Oklahoma were described by Kissel, Dilkes & Reisz (2002) as belonging to Captorhinus magnus, showing the putative tripartite structure visible only from the dorsal surface of the bones. However, this feature was discussed by Rieppel (1993) who argued that the putative unclosed sutures should be interpreted as areas of muscular attachment, or grooves for blood vessel irrigation, or fractures.

Concerning Captorhinus, most of the isolated astragali figured by Peabody (1951) clearly belong to mature animals, according to their size and structure (see Fox & Bowman, 1966, for comparison); the smallest one already shows the same morphology seen in the larger ones. If the astragali shown by Peabody (1951) partially represent an ontogenetic transformation series, they cannot confidently demonstrate that the apparent tripartite structure is derived from the fusion of three or four of the plesiomorphic tarsal bones. A feature that could not support the hypothesis of the tripartite structure is that the sutural lines and groove patterns present in Captorhinus as described by Peabody (1951) are only visible on the ventral surface of the bone; alternatively, it suggests that the fusion started on the dorsal surface and was not completed in adult individuals. The same condition can be observed in the large captorhinid Captorhinus magnus (Kissel, Dilkes & Reisz, 2002).

Fragmentary pedes referred to juvenile and adult individuals of the giant, largest known captorhinid Moradisaurus grandis from the Upper Permian of Niger, were figured and described by O’Keefe et al. (2005) and O’Keefe et al. (2006). Even though the bones were found in association and it was possible to recognize the identity of some of them, they represent isolated and disarticulated pedes whose referral to Moradisaurus can be possible but not accurate, at least no more, than to any other basal tetrapod of the same size. Nevertheless, based on the pes assigned to a juvenile captorhinid, O’Keefe et al. (2006) suggested that the c3 is also a component of the multipartite amniote astragalus, occupying its latero-distal corner. However, the individualization of the constituent bones of the juvenile tarsus and all the possible arrangements show that there is a bone, dorsal to the intermedium that does not belong to the tarsus, unless it is part of the intermedium yet not totally ossified because the juvenile condition of the specimen. However that bone is the only that is totally isolated from the rest of the tarsus, which excepting the four distal tarsals, appears as a co-ossified structure.

Even though our reconstruction of the non-hatched Mesosaurus tarsus is consistent with the O’Keefe et al. (2006) reconstruction of the Moradisaurus tarsus in the fact that the c3 may be part of the astragalus, the arrangement of the bones seems to have been very different in both taxa. Moreover, the putative calcaneum has a very developed notch for the perforating artery, which does not match with the condition in the astragalus, including the evident individualization of the constituent bones. It is also difficult to include the O’Keefe et al. (2006) specimen because their reconstruction does not show an evident fusion between the tibiale and the c4, and because it is a unique, isolated, putatively juvenile pes of Moradisaurus, where the identity of the bones is highly subjective. The other fragmentary pes, interpreted to pertain to an adult specimen displays the typical amniotic tarsal structure and the astragalus shows no sign of the composite origin.

The presumable “implicit” relationship between mesosaurids and basal synapsids regarding the structure of their skull and tarsus

Huene (1941) proposed for the first time a phylogenetic relationship between Mesosaurus and some basal ‘pelycosaurs’. That suggestion was not generally acknowledged by later authors who developed the currently accepted hypothesis that mesosaurids are the basalmost sauropsids (Laurin & Reisz, 1995) or the basalmost parareptiles (Modesto, 1999). More recently, Piñeiro (2004) found some evidence that she understood gave support to Huene’s hypothesis (1941) but acknowledged that it should be tested in a phylogenetic context. Moreover, the nature of the mesosaurid skull, discussed during more than a hundred years, has been recently reassessed to show the presence of a synapsid-like lower temporal fenestra in Mesosaurus tenuidens (Piñeiro et al., 2012c). This contribution gave credit to the observations made by Huene (1941) about the morphology of the mesosaur skull. Similarly, the tarsus of mesosaurs has been studied by several authors, and here we have demonstrated that its structure is almost identical to that described for basal synapsids, but also it is equivalent to that of basal sauropsids, including one of the basalmost diapsid Petrolacosaurus kansensis (Reisz, 1981).

Basal synapsids show a greater development of the calcaneum (Romer & Price, 1940), which in some taxa roughly acquires the size of the astragalus. In contrast to this, the calcaneum of Mesosaurus is smaller than the astragalus (although the size differences are less significant in adult individuals), and develops a lateral expansion in the area of the heel, possibly for insertion of extensor tendons including the Achilles tendon (Fig. 7).

Indeed, the tarsus in early amniotes is both structural and morphologically equivalent in the two groups, except that in ‘pelycosaurs’ there is no evidence for the multipartite formation of the astragalus, thus generating doubts about the homology of these bones in synapsid and sauropsid amniotes (Rieppel, 1993). However, the multipartite original structure can be seen just in very young mesosaurs and it disappears before the achievement of the adult stage and there are few examples of young pelycosaurs individuals for comparative purposes. The composite structure seems to be evident in captorhinids, being possibly an heterochronic pattern.

Evolutionary paths for the development of amniote tarsus: the mesosaur contribution

The morphological ontogenetic transformation presented here for Mesosaurus tenuidens is the most complete known for a basal amniote (cf. Laurin & Reisz, 1995) and as such, it constitutes a relevant database for studies of a different nature. The information provided for this data base on the origin of the amniotic tarsus suggests that, as Peabody (1951) and previous authors (e.g., Holmgren, 1933) have stated, the earliest astragalus originated from at least four ossification centers (taking into account that the tibiale and c4 fuse together early in the ontogeny), near the tibial and fibular distal margins.

According to our observations of the non-hatched Mesosaurus tenuidens which possesses an astragalus formed by at least four bones, we can say that the mesosaurid astragalus is not a neomorphic as Rieppel (1993) has suggested, unless we consider that once united in the earliest stages of the development, these bones form a new element. Even the evidence taken from taxa such as the embolomere Proterogyrinus scheelei Romer, 1970 can provide support for the multipartite hypothesis and the identification of the bones provided in the present work (Holmes, 1984).

We made several interesting observations that support the already established homologies and possible evolutionary paths towards the origin of the primitive amniotic astragalus. Particularly in Proterogyrinus the intermedium has a very similar structure to that of the astragalus of young mature mesosaurs, and the tibiale is clearly sutured against the medial corner formed by c4 and the intermedium. The fibulare is also very similar to the calcaneum of the same stage (see Figs. 1–6), so it is logical to presume that these bones are homologous, as already stated. The main question is what happens to the remaining bones to obtain the mesosaurid (=basal amniote) tarsus consisting of two large proximal elements plus one or two centralia and five distal tarsals. We find evidence for the presence of c3 early in the ontogeny (Fig. 9); it is possible that it fuses to c4 in the described mesosaur ontogenetic transformation after the c4 fuses to the tibiale. Indeed, based on the structure shown by Proterogyrinus (Holmes, 1984), where apparently the tibiale fuses to c4, and taking into account that shown by the tarsus in the captorhinomorph Labidosaurus (Williston, 1917) where the intermedium and the tibiale also fuse to c4, we hypothesized three possibilities or combinations: A, the astragalus is just formed by the intermedium + tibiale only, and c4 and c3 undergo a reduction in size until they finally disappear (not plausible, given the probable presence of c4 and c3 in the tarsus of the non-hatched mesosaurid); B, the astragalus is formed by intermedium + tibiale + c4, and c3 is reduced to be lost (not probable given its putative presence in the tarsus of the non-hatched mesosaurid and taking into account the proposal by O’Keefe et al. (2006)); C, the astragalus results from the fusion of all bones, i + te + c4 + c3 (Figs. 9 and 12A). The last possibility (C) seems to be supported by the materials that we described here, and is consistent with that suggested by O’Keefe et al. (2006), who provided evidence for the inclusion of c3. It does not imply the loss of bones but a re-patterning to produce the amniotic tarsus. Moreover, there are also two possibilities for the formation of the ‘navicular’: 1, it results from fusion of c1 and c2; 2, it is formed by the c2 after the reduction and loss of c1 (see Fig. 12B). We found probable evidence of some of these fusions (the tibiale + ?c4 + intermedium, c1 + c2) in early stages of Mesosaurus tenuidens’s development.

If the hypotheses of the astragalus and the ‘navicular’ formation are combined, we can have the following six possibilities: A-1; A-2; B-1; B-2; C-1; C-2, but the evidence from mesosaurs might support just C-1.

Conclusions

The changes produced in the mesosaur tarsus structure during ontogeny were established based on the study of several specimens preserved in different stages of development. This transformation series is the most complete known for a basal amniote as it includes even embryological information. Our results allow for a better recognition of intraspecific (ontogenetic) from interspecific variation in mesosaurs and provides more informative characters that can be used in comparative studies of amniote relationships.

The mesosaur tarsus includes eight or nine bones: astragalus and calcaneum plus centralia 1 and 2 (fused to form the mesosaur ‘navicular’) and five distal tarsals. The ‘navicular’ is proved to be present in all subadult and adult mesosaurs, even in Mesosaurus where it fuses to the astragalus in mature individuals. The early amniote astragalus is a composite bone as can be evidenced by the presence of at most three sutured bones in the tarsus of a non-hatched mesosaurid in an advanced stage of development. These bones seem to be the intermedium and the tibiale, and the later fused to c4; and the c3. Thus, our study rejects the hypothesis that the amniotic astragalus is neomorphic.

Regarding the analyzed ontogenetic series, we could determine that the attainment of maturity in mesosaurs can be related to a determinate tarsus structure, which can be a good age indicator to extrapolate to other groups of basal amniotes. Moreover, the morphological changes observed in the mesosaur ontogeny could have strong implications in the recognition of until now undocumented, ancestral developmental features of early amniotes.

Current morphological and comparative studies on the mesosaurid skeleton suggest other interesting similarities between mesosaurids and basal synapsids that will be properly described in a forthcoming paper. However, these features are also shared with other basal sauropsids and taxa that are not even amniotes. For instance, mesosaurs share characters with taxa previously known to be closer to Amniota (Panchen & Smithson, 1988; but see also Smithson et al., 1994) but these hypotheses were not phylogenetically evaluated. These taxa are now considered as stem or crown-tetrapods (Olori, 2015; Marjanović & Laurin, 2015) or their affinities are not yet well defined (e.g., Westlothiana), but they still remain close to the earliest amniotes. This commonly shared morphology among apparently unrelated but very basal taxa reflects the primitive nature of mesosaurids, as already noted by Huene (1941) and other paleontologists. The example of the similar tarsal structure observed in mesosaurids, some microsaurs, basal synapsids and non-amniote tetrapods suggests that the evolution of the astragalus and calcaneum as the most typical bones in the amniotic tarsus could be an acquisition obtained much earlier than when the first recognized amniote appeared and walked on the planet.

Supplemental Information

Supplemental Information 1 Supplemental Materials

Figures of non-hatched Mesosaurus tenuidens. Close view of the feet area. Arrows indicates the position of the sutured astragalus.

Click here for additional data file.

We are indebted to Carl Mehling (Fossil Amphibian, Reptile, and Bird Collections, Division of Paleontology of the American Museum of Natural History) who kindly provided the pictures of specimens revised by Olivier Rieppel in his 1993 paper.

Prof. Ivone Cardoso Gonzalez and Lics. Alejandro Ramos, Marcelha Páez Landim and Igor Fernando Olivera assisted us in the revision of the mesosaurid material housed in the Collection of Departamento de Paleontologia do Instituto de Geociências, Universidade de São Paulo, Brazil. Silvia Villar gave us a big help by allowing us to present the best SEM photographs that could be taken of the non-hatched mesosaur tarsus, which, being a unique specimen preserved as an external mould, could not be separated from the compacted shale that contains it, and neither it could be treated with a golden cover before to be photographed. GP wishes to thank Jorge Ferigolo for having allowed her to meet mesosaurs and for the valuable talks together and the outstanding knowledge that he spread to learning and curious people; he made her understand how much we can know from fossils to reconstruct the life during the past. We thank very much Robin Hewison for kindly accepting to revise the English language. We also want to acknowledge Leandro Gaetano and David Marjanović for their insightful comments, helpful criticisms and editorial remarks that highly improved this manuscript.

Institutional Abbreviations

FC-DP Fossil Vertebrates of Facultad de Ciencias, Montevideo, Uruguay

GP/2E Instituto de Geociências (section Palaeontology), São Paulo University, São Paulo, Brazil

SMF-R Senckenberg-Institut, Frankfurt, Germany, MN: Museu Nacional de Rio de Janeiro, Brazil

AMNH American Museum of Natural History, New York, USA

Additional Information and Declarations

Competing Interests

Author Contributions

Data Availability

Graciela Piñeiro is an Academic Editor for PeerJ.

Graciela Piñeiro conceived and designed the experiments, performed the experiments, analyzed the data, contributed reagents/materials/analysis tools, wrote the paper, prepared figures and/or tables, reviewed drafts of the paper, revision of the collections and analysis of the specimens.

Pablo Núñez Demarco conceived and designed the experiments, performed the experiments, analyzed the data, contributed reagents/materials/analysis tools, wrote the paper, prepared figures and/or tables, reviewed drafts of the paper, analyses and comparative measurements of the specimens.

Melitta D. Meneghel conceived and designed the experiments, analyzed the data, contributed reagents/materials/analysis tools, wrote the paper, reviewed drafts of the paper, comparative anatomical and morpho-funcional studies.

The following information was supplied regarding data availability:

The research in this article did not generate any raw data.

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
