# Peer review of "The ontogenetic transformation of the mesosaurid tarsus: a contribution to the origin of the primitive amniotic astragalus"

_PeerJ, doi:10.7717/peerj.2036_

## Round 0.1 · original submission · Major Revisions

I have now received two reviews of your manuscript, they find the study very interesting, although both identify a number of serious flaws that need your careful attention. Both reviewers pointed out that no phylogenetic context is given in your paper. Please, include a cladogram with the analyzed taxa. Likewise, both reviewers are highly concerned in relation to some of your figures (e.g. Reviewer #1 Fig. 5; reviewer #2 Fig. 2 among others) and I agree in that lack of clarity is a serious problem. You should include better figures so that your inferences can be warranted. A number of issues regarding the literature used have been pointed out by our reviewers, such as the quotes of Smithson, I strongly recommend to check carefully your references. Both annotated pdfs convey many suggestions that will serve to improve the discussion of your results.

I would like to see a revised version of your manuscript that takes a point by point account of the comments of both reviewers.

·

Basic reporting

I have annotated the pdf (including the figures!) with suggestions for improvement, mostly clarifications of language and style – make sure you don’t overlook the spaces in my insertions. There are a few sentences that I don’t understand at all.

The gravest issue is Fig. 5. It is entirely possible, but by no means clear, that the indicated sutures are visible in the SEM image. Indeed, the supposed suture between the intermedium and centrale 4 looks like a crack that continues uninterrupted to the other side of the bone, where it is paralleled by several other cracks in the region between the supposed intermedium and the supposed tibiale. I have to ask for a better figure: higher resolution, perhaps taken from a different angle. Similarly, the photo in Fig. 4A has too low resolution to show the sutures that are indicated in the drawing; instead of two sutures, I can only see one structure that may be a suture or a crack. At present, the most newsworthy evidence in the manuscript is really weak.

Experimental design

no comments

Validity of the findings

see above and general comments

Additional comments

This is a very interesting manuscript that tackles several important questions, some very old, some heretofore mostly overlooked. Even disregarding all that, the previously unknown data on the ontogeny of Mesosaurus should definitely be published in PeerJ after appropriate revision; it will be very useful for future studies on the origin and early evolution of Amniota.

I have annotated the pdf (including the figures!) with suggestions for improvement, mostly clarifications of language and style – make sure you don’t overlook the spaces in my insertions. There are a few sentences that I don’t understand at all.

The gravest issue is Fig. 5. It is entirely possible, but by no means clear, that the indicated sutures are visible in the SEM image. Indeed, the supposed suture between the intermedium and centrale 4 looks like a crack that continues uninterrupted to the other side of the bone, where it is paralleled by several other cracks in the region between the supposed intermedium and the supposed tibiale. I have to ask for a better figure: higher resolution, perhaps taken from a different angle. Similarly, the photo in Fig. 4A has too low resolution to show the sutures that are indicated in the drawing; instead of two sutures, I can only see one structure that may be a suture or a crack. At present, the most newsworthy evidence in the manuscript is really weak.

You have overlooked two important recent papers; one only came out a month and a half before you submitted the manuscript, the other three months and a half. They support the hypothesis that the astragalus is a fusion product and not a neomorph by suggesting that this fusion has been undone or partially undone in the clade of large chameleons and in many, perhaps all dinosauromorphs (where the “ascending process of the astragalus” is the intermedium, which has a separate center of ossification and furthermore fuses, in neognath birds, first to the fibulare, not to the tibiale). Thus, they’ll be very useful for your historical review, your statements about current interpretations, and even your interpretation of the ontogeny of Mesosaurus. They do mean that large parts of your manuscript need to be rewritten:
Diaz RE Jr, Trainor PA. 2015. Hand/foot splitting and the ‘re-evolution’ of mesopodial skeletal elements during the evolution and radiation of chameleons. BMC Evolutionary Biology 15:184. DOI 10.1186/s12862-015-0464-4
Ossa-Fuentes L, Mpodozis J, Vargas AO. 2015. Bird embryos uncover homology and evolution of the dinosaur ankle. Nature Communications 6:8902. DOI 10.1038/ncomms9902

Given the evolutionary topic of the manuscript, I’m surprised you didn’t make the phylogenetic context more explicit. You call all tetrapods other than amniotes “amphibians”, which is misleading (some of the ones you mention are actual stem-amphibians, some lie on the amniote stem, and yet others are stem-tetrapods), and you repeatedly call Proterogyrinus, Gephyrostegus and Westlothiana embolomeres – Proterogyrinus is indeed an embolomere, but Westlothiana has never been considered one, and Gephyrostegus hasn’t been in a long time! Gephyrostegus is instead closer to Amniota (and probably Amphibia) than Embolomeri is, and Westlothiana is a lepospondyl just like Tuditanus and Pantylus – likely a stem-amphibian (depending on where Lissamphibia belongs).

Why do you bring up Westlothiana at all? I think you’re using the preliminary conclusions of Smithson (1989), but erroneously cite Smithson et al. (1994) for them and haven’t noticed what the latter paper actually says about this topic, even though you use it for others. Smithson et al. (1994 – originally planned for 1993, but only published in 1994, as the paper actually states) stressed again and again that – contrary to earlier interpretations, including Smithson (1989)! – Westlothiana has the full set of separate proximal tarsals, as reconstructed in their fig. 20A (across the page from fig. 19M, which you have very much used, see below). Your depiction of the tarsus of Westlothiana in Fig. 6 is simply wrong.

Further, as shown in a color-coded version by Meyer & Anderson (2013: fig. 1), in Proterogyrinus, the tibiale and centrale 4 suture to each other, while the tibiale and the intermedium are not even in contact, because c4 instead contacts the tibia. Smithson et al. (1994: fig. 19M) interpreted Gephyrostegus the same way; given the size of the distal end of the tibia, however, I’m inclined to agree with Carroll (1970: 283) and Meyer & Anderson (2013: 488, fig. 1) that the tibiale and the intermedium fuse, while c4 (or c4+c3) lies distal to both. (The “tibiale” identified by Smithson et al. would then be c1, and “c1” would be c2.) If so, these two animals approached the astragalus from opposite directions – their conditions cannot be homologous to each other and cannot be part of a single evolutionary transformation series.

Importantly, the hypothesis that amniotes and lepospondyls are very close to each other (so that Lepospondyli – or Amphibia – is the sister-group of Amniota + Diadectomorpha, and even the seymouriamorphs lie more rootward) is by no means limited to Laurin & Reisz (1997)! It is consensus now, having been found by different people using very different matrices over the last 10 years. The latest work on tetrapod phylogeny (with some amount of review) is the following preprint, which I have to recommend along with several references in it:
Marjanović D, Laurin M. 2015. Reevaluation of the largest published morphological data matrix for phylogenetic analysis of Paleozoic limbed vertebrates. PeerJ PrePrints 3:e1995. DOI 10.7287/peerj.preprints.1596v1
Character TAR 2 in that preprint tracks the number of separate proximal tarsals.

Extended comments on Orobates and Hylonomus are in the pdf file.

You should take a look at Dilkes (2015) for confirmation about the tarsus of Acheloma. It won’t change much, except that Dilkes confirms there is no distal pretarsal.
Dilkes D. 2015. Carpus and tarsus of Temnospondyli. Vertebrate Anatomy Morphology Palaeontology 1:51–87. https://ejournals.library.ualberta.ca/index.php/VAMP/article/view/25234

Similarly, there’s probably no need to cite the ancient Peabody (1952) for Petrolacosaurus. I would rather cite the thorough redescription by Reisz (1981), who used half a page (50, 51) to describe the astragalus of Petrolacosaurus and made large drawings (fig. 21A, B).
Reisz RR. A diapsid reptile from the Pennsylvanian of Kansas. Special Publication of the Museum of Natural History, University of Kansas 7:1–74.
I can send you the pdf, and so can Michel.

You should mention that at least one “microsaur”, Hyloplesion, is known to have separate proximal tarsals (Carroll & Gaskill, 1978; Olori, 2015: 54, fig. 37E). With less certainty, Carroll & Gaskill (1978) also reconstructed this condition for Saxonerpeton and Aletrimyti (“Goniorhynchus”).
Olori JC. 2015. Skeletal morphogenesis of Microbrachis and Hyloplesion (Tetrapoda: Lepospondyli), and implications for the developmental patterns of extinct, early tetrapods. PLOS ONE 10:e0128333. DOI 10.1371/journal.pone.0128333

The name Eosuchia is best circular-filed. Originally coined for Youngina alone (as beautifully evident in the title of Broom, 1924!), it quickly came to refer to a multiply paraphyletic wastebasket taxon for diapsids that are neither lepido- nor archosaurs. More recently (Laurin, 1991), it was defined as the name of a clade that would contain almost all diapsids – but it was anchored on Apsisaurus, which has turned out to be a varanopid synapsid (Reisz, Laurin & Marjanović, 2010; Ezcurra, Scheyer & Butler, 2014), so that Eosuchia has become a junior synonym of Amniota…
Ezcurra MD, Scheyer TM, Butler RJ. 2014. The origin and early evolution of Sauria: reassessing the Permian saurian fossil record and the timing of the crocodile-lizard divergence. PLOS ONE 9:e89165. DOI 10.1371/journal.pone.0089165
Laurin M. 1991. The osteology of a Lower Permian eosuchian from Texas and a review of diapsid phylogeny. Zoological Journal of the Linnean Society 101:59–95.
Reisz RR, Laurin M, Marjanović D. 2010. Apsisaurus witteri from the Lower Permian of Texas: yet another small varanopid synapsid, not a diapsid. Journal of Vertebrate Paleontology 30:1628–1631.

Finally, you use “foetus” (British spelling) and “fetus” (American spelling) interchangeably. You should probably pick one.

David Marjanović

·

Basic reporting

I have some comments and suggestions that are to be found in the pdf file attached; the more important of them follow:
1) I think it would benefit from a fair amount of editing and re-wording. I have made some suggestions in this sense but the opinion of a native English speaker will substantially improve the manuscript readability.
2) The arguments in the discussion are not always easy to follow. I think that the authors should consider revising and re-phrasing the discussion so that the arguments are presented in a clearer way.
3) I think that the discussion of the transition between the basal tetrapod to the amniote astragalus and tarsal structure in a phylogenetic framework would really improve the impact of the manuscript. The authors should consider including a cladogram in order to show the phylogenetic position of the taxa and the different stages in the evolution of the astragalus.
4) The authors should employ widely recognized clades or specify which taxa they are referring to in every case.
5) It is important that the authors explain how do they determine that an individual is a juvenile or an adult and how do they recognize maturity. They introduce “age terminology” but never state the support for such hypotheses. This should be discussed in the methodology section.
6) I am not sure about the usage of the term “fetus”. Maybe the authors should consider replacing it by “non-hatched individual”.
7) O´Keefe et al., 2006 is, in many opportunities along the manuscript, cited as proposing a tripartite origin of the astragalus even though these authors suggest that four ossification centers are involved. I believe that Peabody is also wrongly cited in one occasion. The authors should be careful with these mistakes and check the references.
8) Figure 1 and 2: The lettering is not reader-friendly, a more “traditional” method would do better. The tarsal elements end up being too small and it is impossible to recognize their morphological traits. The authors should split the figure and enlarge the specimens or provide close-ups of each of them in a new figure. Additionally, it is important that they represent and label the structures described and discussed in the text.
9) Figure 3: I think this image is too small. The structures described and discussed in the text should be represented and labeled. Additionally, not all the structures mentioned can be observed in the photographs.

Experimental design

I have some comments and suggestions that are to be found in the pdf file attached; the more important of them follow:
1) I find it troublesome that there are several comparisons of the Mesosaurus tarsus with that of the early synapsids but not with early sauropsids without providing an explanation for this differential treatment.
2) I agree with the authors about the difficulty in identifying O’Keefe et al juvenile specimen as Moradisaurus grandis from the information provided in their paper of 2006; nevertheless, I believe it is not a good idea to just disregard it. The authors disregard O´Keefe et al specimen stating that they cannot find any character to support a precise taxonomic assignation, although they have not analyzed the specimen personally. Thus, until they analyze the specimen personally, the authors should follow O´Keefe et al taxonomic opinion. It is important to note that the same argument could be presented in the future for some of the specimens that the authors confidently assign to Mesosaurus tenuidens (as they do not provide the basis for their taxonomic identifications). Anyway, I strongly recommend the authors to include O’Keefe et al specimen in the discussion as it represents one of the few juvenile basal amniotes with a composite astragalus.

Validity of the findings

I have some comments and suggestions that are to be found in the pdf file attached; the more important of them follow:
1) There are many statements which are not supported by the evidence or references provided. Proper justifications and discussion must accompany all statements particularly those related to homology hypotheses or applicable to a wide array of taxa.
2) The authors acknowledge that the astragalus of Mesosaurus is homologous to the intermedium, tibiale and centrale 4 but they do not justify this affirmation properly. They must include an explanation for their decision before making such statements as the astragalus homologies are still debated.

Additional comments

In this manuscript, the authors analyze the tarsal structure, particularly the astragalus, of several specimens of Mesosaurus tenuidens representing many ontogenetic stages. Additionally, they discuss the astragalus homologies between amniotes and non-amniotes and comments on the evolution of the tarsus. In my opinion, this manuscript is very interesting and worth of publication but it will benefit from some changes. I have some comments and suggestions that are to be found in the pdf file attached. My identity can be revealed to the authors. Please, do not hesitate in contacting me should any question arise.

---

## Round 0.2 · Major Revisions

I think that the ms has been remarkably improved however, it is still not ready for publication. Both reviewers have pointed out a number of issues that require further attention. My main concern relates with the problems of your figures. Doubts about your identification of the tarsal structures persist (see for example Dr. Gaetano's comment on page 15, ln. 233, and general comments of Dr. Marjanović). Considering that your inferences must be warranted by your data, they must be available through the figures. Apart from that, quotes should be checked, and your presentation of Westlothiana deserves further considerations. I would recommend you to clearly state your reasons for your decision regarding the perspectives of Carroll (1970) and Meyer & Anderson (2013) on the one hand and by Smithson et al. (1994) on the other. The issues with the cites of Smithson (1989) and Smithson et al. (1999) persist, could you please explain your preferences here?

·

Basic reporting

Figures 8A and 9B continue to have insufficient resolution to show what they are claimed to show. They should be replaced.

Experimental design

no comments

Validity of the findings

see above (Figs. 8A, 9B) and below

Additional comments

The manuscript has improved, I still recommend publication after rather minor revision. Unfortunately, two important points have not improved.

One of them is the presentation of Westlothiana. The rebuttal letter makes clear that, even though you cite Smithson et al. (1994), you have pretty much exclusively used the earlier description by Smithson (1989). If you have a reason to think that the 1994 paper is wrong and the 1989 paper right about the tarsus, as you imply in line 171, you must spell that reason out. In lines 519–541 you try to argue that the interpretation of 1994 is as uncertain as that of 1989 (which is not the same as arguing that the 1989 version is better); but in the whole rest of your paper you simply take for granted that the 1989 version is correct, even citing (in line 512) the 1994 paper when you mean the 1989 paper alone. In lines 527–528 you state that “a proposal about the tarsus structure in Westlothiana would be very speculative at this stage”, but in the whole rest of the paper you simply take for granted that Westlothiana had an astragalus! At the very least, the manuscript cannot remain so self-contradictory.

The other is the quality of the photos of FC-DPV 1502 (fig. 8A) and 2504 (fig. 9A). Even in the new PNG pictures, the resolution is too low to clearly show the sutures. I can only start from the drawings and then try to interpret the lines from the drawings into the photos – and even this doesn’t work well at all. Fig. 9A in particular is just blurry, and the new hand-drawn outlines make it altogether impossible to see if sutures are hidden under them! I have to insist that everything shown in the drawings must be reasonably clearly visible in the photos themselves. The point of a descriptive paper is to make sure that the readers will not need to see the specimens themselves – of course we’ll never quite get there, but we have to make a serious effort toward this goal. Please take new pictures. – In addition, it would be very interesting to see the separated proximal tarsals in the other foot of FC-DPV 2504 (mentioned in line 193)! I’m surprised you didn’t include a photo of it.

Concerning the separate intermedium in birds, keep in mind that the “ascending process of the astragalus” is found all the way to the origin of Dinosauromorpha (even Lagerpeton has it). While the undoing of the astragalus in certain chameleons is clearly an adaptation to perching, the undoing of the astragalus in dinosauromorphs cannot possibly be an adaptation to flying (or to perching for that matter); note that Ossa-Fuentes, Mpodozis & Vargas (2015: 4) explicitly state that the ornithischian Fruitadens has a suture between the “astragalus” and the “ascending process”! I take this as further confirmation of the hypothesis that the astragalus is composite; usually its components fuse very early in amniotes, but they still arise separately and can fuse later or not at all.

In the phylogeny part of the discussion, you cite a whole bunch of papers for things they never said. It’s stunning. Details below.

Abstract:
“the astragalus, clearly formed by the suturing of three bones, which we interpret as being the intermedium, the tibiale, probably already fused to c4 and the c3” is confusing; do you mean “the intermedium, the tibiale, and c4, to which c3 may already be fused”?
“A primitive, amniote-like tarsal structure is also observed in” – remove “primitive,”; it sounds as if the amniote-like structure were plesiomorphic for tetrapods. “Amniote-like” is already clear enough. See above on Westlothiana, which does not belong in this sentence at all.
In the last sentence, replace “are not” by “may not be”, and “they might” by “but may instead”.

This time I cannot annotate the pdf, so I provide my detailed comments below.

Lines 39–44: These two sentences contradict each other. The first says Holmgren (1933) and Peabody (1951) supported the origin of the astragalus from the intermedium alone, or perhaps from intermedium + tibiale, while the second says that these same papers instead supported the origin of the astragalus from intermedium + c4 or perhaps intermedium + c4 + tibiale. Please clarify.
43: Remove the last comma, or add another one before “perhaps”.
47: Please explain why you agree with Schaeffer (1941) and Holmes (1984) rather than with Carroll (1970) and Meyer & Anderson (2013).
48: “which possesses an amniote-like tarsus” is redundant with the statement that there is fusion in the tarsus. I would drop it to make for more fluid reading.
51: Remove the second comma.
55: Hylonomus (Meyer & Anderson, 2013) is not a captorhinid. It’s probably closer to Diapsida than to Captorhinidae (Müller & Reisz, 2006).
56: “prove” sounds so mathematical; replace it by “show” or “demonstrate”.
59–60: Replace by “and therefore the presence of an additional anlage for the tibiale remains contentious”.
61: Remove the comma.
81: Insert a comma behind “stage”, and insert “or hatchlings” after “newborns”.
83: Replace “can be” by “is” – “possibility” is clear enough.
87: Replace “Geociencias” by “Geociências”.
106: Insert a comma behind “tarsus”.
109: Insert a comma behind “development”.
118: Replace “arch” by “arches”.
121: Replace “characterizing” by “distinguishing”.
125: Replace “at” by “in”, and remove the comma.
126: Replace “the stylized” by “their final” or “a more complex” or suchlike.
128: Remove “it”.
129: Replace “specimens” by “individuals”.
135: Remove the comma.
139–140: Remove “in the tarsus,”.
148: Insert “as” behind “even”.
151: Replace “to” by “for”.
155: Insert “to” behind “refer”.
156–157: Replace “used” by “use”, and format the quotation marks (deleting and rewriting them should do the trick).
161: Remove the first comma.
163: Which is it, “1864—1865” or “1865”? Only one publication can be the one where the name was first validly published.
169, 170: Tuditanus and Pantylus were not named in these publications, but a hundred years earlier! I recommend: “(particularly Tuditanus and Pantylus; Carroll & Baird, 1968; Carroll, 1968)”.
171: see above.
180, 246, 252, 270: Remove the first hyphen. It’s just confusing.
184: Replace “for” by “by”.
191: Insert a space in front of “9”.
196: Replace “zeugopodial,” by “zeugopodium”; remove both commas, and replace “were weathered” by “decomposed”.
200: Replace “avoided” by “prevented”.
202: Remove “the precursor,”, and replace “fussed” by “fused”!
211: Replace “could not yet be ossified” by “could be still unossified”, because “could not yet be ossified” would be interpreted as “it is impossible that they were already ossified”. – Why “but”? In the rest of the sentence you go on to state that this (ontogenetic lack of ossification) is exactly the case, if I understand it correctly.
216, 234: Replace the first hyphen by a space to avoid confusion.
223: Simply remove “(calcified”.
229: Replace “can more possibly be” by “are more likely”.
262: Insert a space before “3”.
265: Remove the space from “midlength”.
276: Insert a space before “6”.
289: Replace “experiences” by “experienced”.
293: Remove “post birth,”, and insert “postnatal” in front of “mesosaur”.
295: I don’t understand “and the one mostly preserved”. Please reword.
298: Remove the last comma.
301: Remove “both” – it implies emphasis, and there’s nothing to emphasize there.
303: Here, in contrast, such emphasis would make sense. I suggest: “artery is not visible. We consider both of these features useful in identifying juvenile, immature”. This would work best if you replace the period in line 302 by a semicolon (;).
320: Remove “bone”, and remove “and” at the end of the line.
329: Replace “limbs” by “limb”.
337: Replace either “former” by “first” or “second” by “latter”.
340–341: Replace “proven to occur in” by just “from”.
347–348: Reword after the comma: “the fibulare (calcaneum) ossifies much later in mesosaurs and Hovasaurus.”
357: Replace “later, to the intermedium” by “to the intermedium later”.
362: Replace “, and” by “;”.
374: Replace “It” by “This bone”.
390: Replace “deduct” by “deduce”.
391–392: Reword: “hatched mesosaurid (Fig. 9) where we interpret the only tarsal bone as the astragalus, with sutures indicating the intermedium, the tibiale and maybe the”.
395: Remove the space from “mediodistal”.
404: Remove “it”.
421: Replace “firstly” by “first”.
424: Remove “to”.
430–431: Reword after the comma: “but this needs to be tested by analysis of more than one individual of the same species at different stages of development.”
447: Remove the misleading comma.
450: Replace “stages” by “stage”.
454: Replace “calcaneous” by “calcaneum”.
460: Replace “despite” by “because”…!
460–463: see above.
468: Move the comma to before “and”.
473–476: These are simply not “closely related”. Within the smallest clade formed by all of these, according to your references and all other publications since the mid-20th century that I can think of, lie the seymouriamorphs, which lack any tarsal fusion, Microbrachis, which also lacks it, and the diadectomorphs Limnoscelis and Tseajaia which lack it as well (note that Tseajaia is the sister-group of Diadectidae).
475: Why cite Ruta, Coates & Quicke (2003) instead of the enlarged and (slightly) updated version by Ruta & Coates (2007)? – You managed to put the accent on my c in line 430; please do it again. :-) However, neither the 2007 nor the 2008 paper are at all relevant here; the 2009 paper and the 2015 reprint are! – Replace “for us to assert” by “for our assessment of”, except that that’s not what you did, see above.
478: Remove the misleading comma, and replace “possible” by “possibly”.
480–482: As I wrote last time, this is not what Carroll (1970: 283) wrote. Instead, he wrote: “In Gephyrostegus, the tibiale and intermedium are fused dorsally, although the originally separate centres of ossification are represented by radiating patterns of fine striations. Ventrally the bones are suturally united, but not fused. The proximal centrale remains an independent centre of ossification, but articulates closely with the intermedium and tibiale.” In short, the c4 is not fused to the tibiale! There is a genuine conflict between the interpretations by Carroll (1970) and Meyer & Anderson (2013) on the one hand and by Smithson et al. (1994: fig. 19M) on the other. I have explained why I prefer the former interpretation; if you prefer the latter, you need to explain why. Right now you’re acting as if the conflict isn’t there; it is there.
484: Replace “1972” by “1968”. – Why “but see also”? Carroll & Gaskill (1978) continued to assert that Tuditanus has an astragalus. This time, there really is no conflict. (By the way, I have seen that astragalus myself. It is real.)
484–485: No, see above (Smithson et al., 1994).
492: Which relationships exactly do you mean?
494–495: Again, our 2007 paper treats a completely different topic, and Gephyrostegus and the anthracosaurs are not included in our 2008 analysis. Even in the 2009 paper the taxon sample is too small to be cited here.
495: Not “most”. All of them. I’m not aware of any paper that has found Proterogyrinus to be anything other than an embolomere.
496: Replace “anthrachosaur” by “anthracosaur”.
496–497: The relationships of Proterogyrinus are unresolved within Embolomeri. The uncertainty does not extend any farther, and it is wrong to cite Ruta, Coates & Quicke (2003) or any other publication I know for such a claim.
497–500: This hypothesis isn’t “accepted” by “researchers”, but found by analyses; except it’s not always – in some trees from Marjanović & Laurin (2015), Temnospondyli is closer to the seymouriamorphs and microsaurs (and amniotes) than Gephyrostegus is. Besides, Michel is a coauthor on all the cited papers, so “other researchers” is only partially true anyway.
501–502: This is a very, very surprising claim. Not one of these papers have suggested microsaur paraphyly with respect to amniotes. Laurin (1998) and its update by Vallin & Laurin (2004) found microsaur paraphyly with respect to Lissamphibia, not with respect to Amniota; this puts the microsaurs on the amphibian stem, not on the amniote stem. Anderson (2001) likewise found microsaur paraphyly with respect to the only lissamphibian included in the analysis, Eocaecilia; no amniotes were included in the analysis, and the only included diadectomorph fell outside of Lepospondyli. Olori (2015) did not perform a phylogenetic analysis at all, but accepted the temnospondyl hypothesis on lissamphibian origins, according to which Lepospondyli is a clade of stem-amniotes simply because it lies, as everyone agrees, closer to Amniota than to Temnospondyli. Seriously, what a mess. It really looks like you haven’t read any of these papers before citing them. – Finally, as I mentioned last time, Olori (2015) confirmed that the microsaur Hyloplesion does not have an astragalus (complete or partial). If you think that Tuditanus and Pantylus are more closely related to Amniota than Hyloplesion is, you need to present an argument for this novel claim; it hasn’t been made anywhere in the literature.
511–512: Not Westlothiana, see above and Smithson et al. (1994), which you cite in line 512 for the opposite of what it says.
513: At least remove “very”. The only amniote-like feature in the tarsus of Proterogyrinus is the fusion of tibiale and c4, and the only amiote-like feature in the tarsus of Gephyrostegus is the fusion of what seem to be tibiale and intermedium.
515: Replace “;)” by “),”, and remove “within”.
533–534: An L-shaped intermedium is actually quite widespread, even though the distribution is rather sporadic; it occurs in Tulerpeton, Caerorhachis, Proterogyrinus, Archeria, and Westlothiana. I think the astragalus is usually L-shaped because it contains the intermedium. Note, though, that not every astragalus is L-shaped; Permo-Carboniferous counterexamples include Pantylus and the amniote Eocasea.
537: That’s actually a rather labile character both outside of Amniota (see character 240, TAR 3, in Marjanović & Laurin, 2015) and within it. (It even occurs as intraspecific variation in some taxa.) The diadectomorphs all retain the contact, incidentally, so its presence wouldn’t be unexpected in a very close relative of Amniota.
542: Remove “and”.
544–549: However, Olori (2015) describes Hyloplesion as a microsaur without an astragalus. It has a separate tibiale, intermedium and fibulare.
547: Why “other”? Do you think microsaurs are amniotes?
555: It is true that Berman (2000) considered the diadectomorphs amniotes; but keep in mind that 1) the non-diadectid diadectomorphs, Limnoscelis and Tseajaia, do not appear to have an astragalus, but seem to retain separate proximal tarsals (see below), and 2) not one of the phylogenetic analyses performed since then has found Diadectomorpha to lie within Amniota.
566: Replace “homologated” by “homologized”.
568: Replace “and” by a semicolon.
573: Remove the comma.
581–582: These is only one known tseajaiid, Tseajaia campi. Reword the whole clumsy sentence: “The non-diadectid diadectomorphs (Tseajaia and Limnoscelis) do not have an astragalus, but we wonder if this due to incomplete preservation” – if you really have evidence for that. The reconstruction of Tseajaia by Moss (1972) looks strange enough that I can’t have confidence in it; but for Limnoscelis such an argument is much harder to make (Kennedy, 2010: 218).
Kennedy NK. 2010. Redescription of the postcranial skeleton of Limnoscelis paludis Williston (Diadectomorpha: Limnoscelidae) from the upper Pennsylvanian of El Cobre Canyon, northern New Mexico. New Mexico Museum of Natural History and Science Bulletin 49: 211–220.
611: “spell out” is from a comment of mine; in the text “propose” would be better.
613: Explain why “we know”.
617: In that case, you need to explain what it is that Meyer & Anderson interpreted as sutures.
618: Replace “lyelly” by “lyelli”!
629: The closing parenthesis should not be in italics. – Again, “demonstrate” is better than “prove” in science.
641–645: Exactly, it suggests that the fusion started on the dorsal surface, just like in Gephyrostegus. It is very common that bones begin to fuse on one side or one end and only later fuse throughout their contact. There is no evidence here against the hypothesis of the compound origin of the astragalus.
647, 651: Replace “pes” by its plural “pedes”.
652: What would that be, though? The amniote Bunostegos? The temnospondyl Nigerpeton?
655: Replace “to pertain” by “as pertaining”, and add a comma behind “specimen” (which should rather be “individual”).
656: Replace the second “the” by “a”.
663: Replace “theory” by “hypotheses”. A theory is something much larger.
668: Replace “to denote the presence of” by “as showing” or “as containing”.
670: Remove “several years ago”; it is redundant, and it is misleading because “several” never means as much as 71!
673: Replace “also it is” either by “it also is” or by “also that it is”. – Being an araeoscelidid, Petrolacosaurus is probably no more basal than Araeoscelis. I suggest: “one of the basalmost diapsids,”.
680: Replace “structural” by “structurally”.
698: Replace “birds” by “dinosaurs (notably including birds)” to account for the suture in Fruitadens. – Remove the space from “abovementioned”.
699: Remove the comma, or move it to behind the parenthesis.
701: Replace “Diaz & Trainor (2015” by “Diaz & Trainor, 2015”.
703–704: It is correct but entirely misleading to call this “the most common condition in birds”; it is an autapomorphy of Neognathae. The much smaller clade Palaeognathae retains the plesiomorphic condition.
705: Replace “dinosaurs” by “dinosauromorphs”.
706: Remove the period.
710–712: Reword the whole clumsy sentence as: “However, although centralia are present in squamates and inferred to be basic components of the early amniote astragalus, have not been found in dinosauromorphs, chameleons, crocodiles or turtles.” (I infer “crocodiles or turtles” from “many of [sic] extant diapsids” and “outgroup lepidosauria”.)
713–714: I don’t understand “of the stages of their ontogeny”.
715: I’m unhappy with “advanced”; replace “most of the advanced taxa” by “the abovementioned clades”.
715–717: This sentence lacks a finite verb. Please repair it.
717–719: Reword the sentence as: “Thus, the acquisition of unitary, stout structures instead of several separate, delicate bones was an improvement for fast, sustained locomotion.” That isn’t what chameleons do, so it is not a surprise that they don’t need an astragalus.
722: Replace “neomorph” by “neomorphic”, and remove the second comma.
735: The intermedium is not sutured to the tibiale in Proterogyrinus. It doesn’t even contact the tibiale; instead, as is common in such basal tetrapods, the c4 contacts the tibia! I already mentioned this last time.
739: Remove the question mark, and replace “about” by “for”.
740: Replace the comma by a semicolon.
741: “the latter” would be the transformation; replace “the latter” by “c4”.
747: Remove the space before the closing parenthesis.
750: Move “proposal” to the beginning of the line, and add “by” after it.
752: Replace “to” by “with”, and “which” by “who”.
757–758: “the development of Mesosaurus tenuidens” would be less awkward.
776–777: reword as “the intermedium and the tibiale, the latter fused to c4 and c3” – if that’s what you mean.
778: Has it actually ever been suggested that the calcaneum is a neomorph?
788: Insert a semicolon behind “1988”.
789: Insert a comma behind the parenthesis.
790: Replace “stem tetrapoda or crown tetrapoda” by “stem- or crown-tetrapods”.
791: The phylogenetic position of Westlothiana is very well defined: the latest analyses (Vallin & Laurin, 2004; Ruta & Coates, 2007; Marjanović & Laurin, 2015) have all found it as a lepospondyl, the sister-group to all other lepospondyls – close to Amniota, but less close than Diadectomorpha and not closer than Microbrachis or Hyloplesion.
795: Remove the space from “non-amniote”.
797–798: In this case you need to postulate several early reversals, as mentioned above.
805: Remove “who”.
806: Replace “Geociencias” by “Geociências”.
861: Remove the space from “d’Histoire”, and replace “Séries” by “série”.
897: “Abteilung A” (‘division A’) should be all in italics or all not in italics; or shorten it to just “A” (not in italics).
902: Replace “Science” by “Sciences”.
915: Remove “5729”!
916: “Anatomy” needs a capital letter.
961: “Communications” needs a capital letter.
977: There shouldn’t be a space in “Konservat-Lagerstätte”; please check if the published paper has one.
1028: Replace “Societe Imperiale” by “Société Impériale”.
1048: I bet the apostrophe doesn’t belong here.
1167, 1192, 1207, 1218, 1223: Insert a comma behind “tenuidens”, and remove “transition”.
1173–1174: Reword the sentence as: “See the interpretive drawings in Figs. 2A and 7 and the text for further description.”
1184: Replace “the” by “a”.
1186: Remove both commas.
1194: Remove “the”, and replace “specimens” by “specimen”.
1197: Replace “after-hatched” by “hatchling” – or indeed “newborn”, given the evidence for ovoviviparity.
1205: Remove the space before the period, and one of the two spaces in front of “pa”.
1256: Replace “tíbia” by “tibia”.
Fig. 10: See above for Gephyrostegus; it would be good to include Hyloplesion.
1272: Acheloma may not be any farther away from Amniota than Embolomeri is; see Laurin & Reisz (1999) and Marjanović & Laurin (2015) for discussion.
1274: Replace “actually primitive amniotes” by “primitive actual amniotes” or, better, by “early amniotes”.
1280: Correct the spellings of Carroll and Westlothiana.
Fig. 11d shows the results from some of our bootstrap trees, not any of the most parsimonious trees; those are all identical to Fig. 11a. Concerning Fig. 11c, replace Laurin & Reisz (1999) by its latest update (Vallin & Laurin, 2004).

David Marjanović

·

Basic reporting

no comments

Experimental design

no comments

Validity of the findings

no comments

Additional comments

In this manuscript, the authors analyze the tarsal structure, particularly the astragalus, of several specimens of Mesosaurus tenuidens representing many ontogenetic stages. Additionally, they discuss the astragalus homologies between amniotes and non-amniotes and comments on the evolution of the tarsus. This is the second time that I review this manuscript and the authors have performed most of the changes I suggested. In my opinion, this manuscript is very interesting and worth of publication with only minor corrections. I have some comments and suggestions that are to be found in the pdf file attached; please, do not hesitate in contacting me should any question arise.
My identity can be revealed to the authors. Please, do not hesitate in contacting me should any question arise.

---

## Round 0.3 · Minor Revisions

The ms has been greatly improved. I find your response to the rebuttal letter acceptable, except for this point: “We used in this case the tree of Laurin and Reisz (1999), and we prefer to keep it. “

As you may know, the cladistic hypotheses tend to be improved by adding data or performing better analyses, this is why most people select the last one. I would like you to explain your reasons to select another one.

I have only minor suggestions in the annotated PDF.

---

## Round 0.4 · accepted · Accept

Thank you for your effort in making the suggested changes. The ms is ready for publication.